# Inhibition mediated by group III metabotropic glutamate receptors regulates habenula activity and defensive behaviors

Anna Maria Ostenrath [1], Nicholas Faturos[1,2], Yağnur Işık Çiftci Çobanoğlu[1,3], Bram Serneels [1], Inyoung Jeong[1,4], Ekin Dongel Dayanc [1,5], Anja Enz[6], Francisca Hinrichsen [1], Aytac Kadir Mutlu [1], Ricarda Bardenhewer[1], Suresh Kumar Jetti[7], Stephan C. F. Neuhauss [6], Nathalie Jurisch-Yaksi [1,4] & Emre Yaksi [1,3] ✉

Inhibition plays a key role in brain functions. While typically linked to GABA, inhibition can be induced by glutamate via metabotropic glutamate receptors (mGluRs). Here, we investigated the role of mGluR-mediated inhibition in the habenula, a conserved, glutamatergic brain hub involved in adaptive and defensive behaviors. We found that zebrafish and mice habenula express group III mGluRs. We showed that group III mGluRs regulate membrane potential and calcium activity of zebrafish habenula. Perturbing group III mGluRs increased sensory-evoked excitation and reduced selectivity. We identified inhibition as the primary communication mode among habenula neurons. Blocking group III mGluRs reduces this inhibition and increases neural synchrony. Consistently, we demonstrated that multisensory integration in the habenula relies on competitive suppression, that partly depends on group III mGluRs. Genetic and pharmacological perturbation of group III mGluRs amplified neural responses and defensive behaviors. Our findings highlight an essential role for mGluR-driven inhibition in encoding information and regulating defensive behaviors.

Inhibition is a crucial part of brain function[1]. Apart from controlling brain excitability[2–6] and stabilizing network activity[7,8] inhibition also contributes to multiple neural computations from refining sensory representations[9–26] to mediating high-order cognitive processes[27–33]. Notably, studies in humans using functional brain imaging have highlighted a decrease in brain activity during attention-demanding cognitive tasks globally, except for the area involved in those specific tasks[34–36]. γ-aminobutyric acid (GABA) is the main inhibitory neurotransmitter of the central nervous system[37], and is released from diverse classes of GABAergic neurons with various roles in neural computations[38–40]. Besides GABA, neuromodulators such as dopamine[41], serotonin[42–45] and acetylcholine[46,47] were also shown to mediate inhibition through their specific inhibitory receptors. Glutamate, which is the main excitatory neurotransmitter of the vertebrate central nervous system[48,49], can also lead to inhibition through the action of group II and group III metabotropic glutamate receptors (mGluRs)[49–51]. Group II/III mGluRs are expressed throughout the brain[50,52–54], and modulate brain excitability by acting on the pre- and

[1]Kavli Institute for Systems Neuroscience and Center for Algorithms in the Cortex, Norwegian University of Science and Technology, Trondheim, Norway. [2]Medical Scientist Training Program, University of Minnesota School of Medicine, Minneapolis, MN, USA. [3]Koç University Research Center for Translational Medicine, Koç University School of Medicine, Istanbul, Turkey. [4]Department of Clinical and Molecular Medicine, Norwegian University of Science and Technology, Trondheim, Norway. [5]Department of Physiology, Institute of Health Sciences, Acibadem Mehmet Ali Aydinlar University, Istanbul, Turkey. [6]University of Zurich, Department of Molecular Life Sciences, Zurich, Switzerland. [7]Neuro-Electronics Research Flanders, Leuven, Belgium. ✉e-mail: emre.yaksi@ntnu.no

post-synapse of neurons as well as glia[50,55,56]. To date, the role of glutamate-driven inhibition in neural computations and animal behavior remains elusive.

In this study, we investigated the role of glutamatergic inhibition in brain function by specifically focusing on the habenula (Hb) and group III mGluRs. The Hb is a diencephalic nucleus associated with adaptive behaviors[57–64] and learning[58,61,63,65–70]. Hb dysfunction in humans is linked to mood disorders[71–73]. Hb receives diverse inputs from cortico-limbic[65,66,74–83] and sensory brain regions[84–87], acting as a hub integrating and relaying information to target monoaminergic nuclei that regulate animals' behavioral states[58–60,62,69,88–96]. The evolutionarily conserved[58,63,68,97] subdivision of the mammalian medial (MHb) and lateral (LHb) Hb, which correspond to the dorsal (dHb) and ventral Hb (vHb) in zebrafish, mediate distinct functions. The vHb/LHb is involved in motivation[98,99], decision making[60,88], learning[58,100], defensive[66,101–103] and coping behaviors[57,92,104]. The dHb/MHb is associated with circadian rhythms[105,106], social behaviors[62], sensory processing[84,105,107–110] and fear learning[59,63,68].

Recent studies have shown that the Hb is composed of molecularly diverse[111–113] and spatially organized[58,63,68,97] clusters or ensembles[114] with distinct developmental origins[97,109,115] and function[58,64,107–110]. These spatially organized clusters of Hb neurons display correlated spontaneous activity among nearby neurons and robust anti-correlations across Hb clusters, as shown in zebrafish[107,109,110,116]. Such spatially structured spontaneous Hb activity is largely driven by the cortico-limbic structures of the forebrain[110]. Besides showing correlated spontaneous activity, neurons in individual Hb clusters also exhibit synchronized responses to odors, light and mechanical stimuli[107,109,110]. Interestingly, odor responses in the Hb were shown to inhibit distinct Hb neurons and dampen the way Hb integrates information from its forebrain inputs[110]. These sensory-evoked decreases in Hb responses and robust anticorrelations in spontaneous Hb activity suggest that inhibition plays an important role in Hb function. However, the nature and function of inhibition in Hb networks, which is largely a glutamatergic brain region[113,117], is not well understood.

In this study, we showed that sensory stimuli evoke both excitation and inhibition in Hb neurons. We identified the expression of several group III mGluRs, which are inhibitory, in both zebrafish and mouse Hb. We observed that pharmacological blockage (and activation) of group III mGluRs can increase (and decrease) membrane potential and calcium signals of zebrafish Hb neurons. Our results revealed prominent inhibitory interactions across Hb neurons, when single Hb neurons are micro-stimulated. We also observed that inhibition is the primary mode of integration in Hb neurons, when different sensory modalities are delivered simultaneously. These inhibitory interactions are reduced by perturbing group III mGluRs, which leads to amplified sensory responses and decreased selectivity in Hb neurons. Finally, we identified that genetic perturbation of mGluR6a, which is highly enriched in the Hb, led to stronger defensive behaviors and reduced adaptation to environmental changes. Altogether, our results reveal that inhibition via group III mGluRs plays an important role in neural connectivity and sensory computations in the Hb and regulates defensive behaviors.

## Results

### Habenula neurons respond to sensory stimulation with excitation and inhibition

To investigate the extent of sensory-evoked inhibition in Hb, we monitored Hb responses to olfactory, visual and mechanical stimulation, by using volumetric two-photon calcium imaging in 3 weeks old juvenile *Tg(elavl3:GCaMP6s)* zebrafish[109,110,118–121] expressing GCaMP6s pan-neuronally. We identified that all three sensory modalities evoked both excitation and inhibition in a fraction of neurons, distributed across dorsal and ventral Hb (Fig. 1, Supplementary Fig. 1). These results demonstrate that the zebrafish Hb responds to multiple sensory modalities both by excitation and inhibition.

### Group III mGluRs are expressed in habenula and mediate inhibition in habenula neurons

A large fraction of vertebrate Hb neurons is glutamatergic[109,113,117]. Moreover, the primary pathways delivering visual[84,85] and olfactory[86,122] information to zebrafish Hb are glutamatergic. Hence, we asked whether inhibitory mGluRs are present in Hb, which could explain part of the sensory-evoked inhibition (Fig. 1) and anticorrelations between Hb neurons during spontaneous activity[107,110]. We specifically focused on group III mGluR, since a previous study reported diverse expression of these receptors across the zebrafish brain[123]. To investigate the group III mGluR expression in vertebrate Hb, we examined and compared two recently published single cell RNA sequencing data from zebrafish[113] and mice[117]. We observed prominent expression of mGluR4, mGluR6a, mGluR7 and mGluR8b in zebrafish Hb, with stronger expression in dHb neurons, when compared to vHb (Fig. 2A). Similarly, mGluR4 and mGluR7 were expressed prominently in mouse Hb, but with stronger preference to LHb neurons (Fig. 2B). In the mouse single cell RNA sequencing data, we could detect only few neurons with mGluR8 expression in LHb, and no expression of mGluR6. Due to low expression levels of GPCRs, detection of mGluR transcripts is difficult with single cell transcriptomic profiling[124]. Encouraged by earlier in situ stains of mGluRs in zebrafish larvae[123], we generated a new spatial transcriptome of zebrafish Hb with subcellular spatial resolution[125–127]. The spatial transcriptomic approach was effective in detecting mGluR transcripts. Notably, we identified topographically distinct expression of mGluR4, mGluR6a, mGluR8b in the zebrafish dHb (Fig. 2C), which is in line with our single cell RNA sequencing results. Horizontal sections across the entire forebrain, revealed that while mGluR4 and mGluR8b are also present in other forebrain regions, mGluR6a expression is relatively specific to zebrafish Hb (Fig. 2D), and the olfactory bulbs (Supplementary Fig. 2A). mGluR6b is not present in Hb but is expressed in the olfactory bulbs (Supplementary Fig. 2B, C). Spatial transcriptomic cross sections of the mouse brain[54] also support the habenular expression of group III mGluRs (Supplementary Fig. 2D, E). These results reveal that group III mGluRs are present in both zebrafish and mouse Hb, with different preferences to distinct Hb subregions.

Next, we asked whether group III mGluRs regulate the membrane potential of dHb neurons in juvenile zebrafish whole-brain explant, by using whole-cell-patch clamp recordings and pharmacological treatments (Fig. E, F). To ensure that pharmacological manipulation of group III mGluRs are specific to dHb neurons and not due to effects on neurons presynaptic to the dHb, we first blocked the synaptic transmission by using cadmium chloride ($CdCl_2$, 100 μM), a blocker of voltage gated calcium channels. Next, in the absence of synaptic inputs, we applied a group III mGluR agonist L-(+)−2-Amino-4-phosphonobutyric acid (L-AP4, 10 μM). We observed a significant decrease of membrane potential in dHb neurons (Fig. 2E), suggesting that group III mGluRs can directly inhibit dHb neurons. In parallel, we also identified that adding (RS)-α-Cyclopropyl-4-phosphonophenylglycine (CPPG, 300 μM), a blocker of group III mGluRs, significantly increased in the membrane potential of dHb neurons (Fig. 2F). These results showed that group III mGluRs directly modulate the membrane potential of dHb neurons.

Subsequently, we investigated the impact of group III mGluR agonists and antagonists across the zebrafish forebrain, by using two-photon calcium imaging in *Tg(elavl3:GCaMP6s-nuclear)* 3-weeks-old juvenile zebrafish whole-brain explant[110,128–130]. Bath application of group III mGluR antagonist CPPG (300 μM) resulted in an increase of calcium signals mainly in Hb and around the anterior forebrain (Fig. 3A). To quantify the distribution of CPPG-activated neurons further, we used anatomical landmarks to delineate forebrain regions[110,131,132] (Fig. 3B). We observed that Hb and the olfactory bulbs are the only forebrain regions with increased calcium signals in a significant fraction of their neurons, with significantly larger fraction in

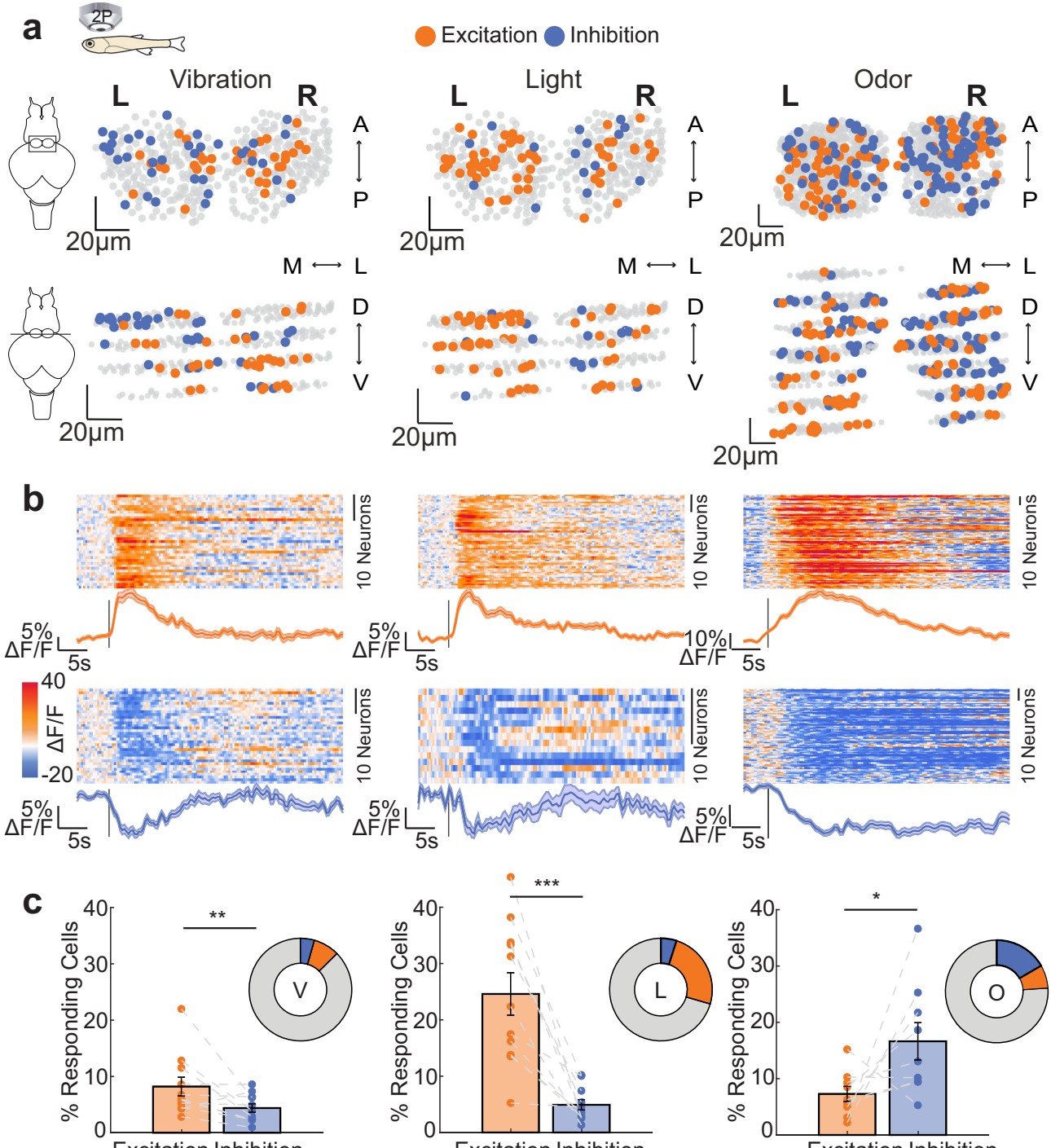

**Fig. 1 | Habenular neurons respond to vibration, light and odors with both excitation and inhibition. a** Representative examples showing the location of responding neurons in three-dimensional Hb reconstruction in *Tg(elavl3:GCaMP6s)* juvenile zebrafish in the dorsal-ventral view (top) and the coronial view (bottom). Neurons are color-coded by their response to vibration (left), light (middle) and odor (amino acid mixture, right). Neurons with activity 2 STD higher than baseline are orange (excitation), 1 STD smaller than baseline are blue (inhibition), non-responding neurons are grey. Scale bar represents 20 μm in x, y and z direction. L left, R right, A anterior, P posterior; D dorsal, V ventral, M-L medial to lateral. **b** Time-courses of habenular calcium signals (ΔF/F) of the excited (top) and inhibited (bottom) neurons from panel (**a**) to the vibration (left), light (middle) and odor (right) stimulations. In the heatmap warm colors indicate excitation, cold colors represent inhibition. Orange and blue colored lines represent average excitation and inhibition, respectively. Stimulus onset is indicated by the black line. Shadow represents +/- SEM. **c** Percentage of excited and inhibited habenular neurons upon vibration (left, *n* = 11 fish, **p = 0.0098), light (middle, *n* = 11 fish, ***p = 0.0005) and odor (right, *n* = 9 fish, *p = 0.0273) stimulations. (one-sided Wilcoxon signed-rank test). Error bar represents mean +/- SEM. Also, see Supplementary Fig. 1. Source data are provided as a Source Data file.

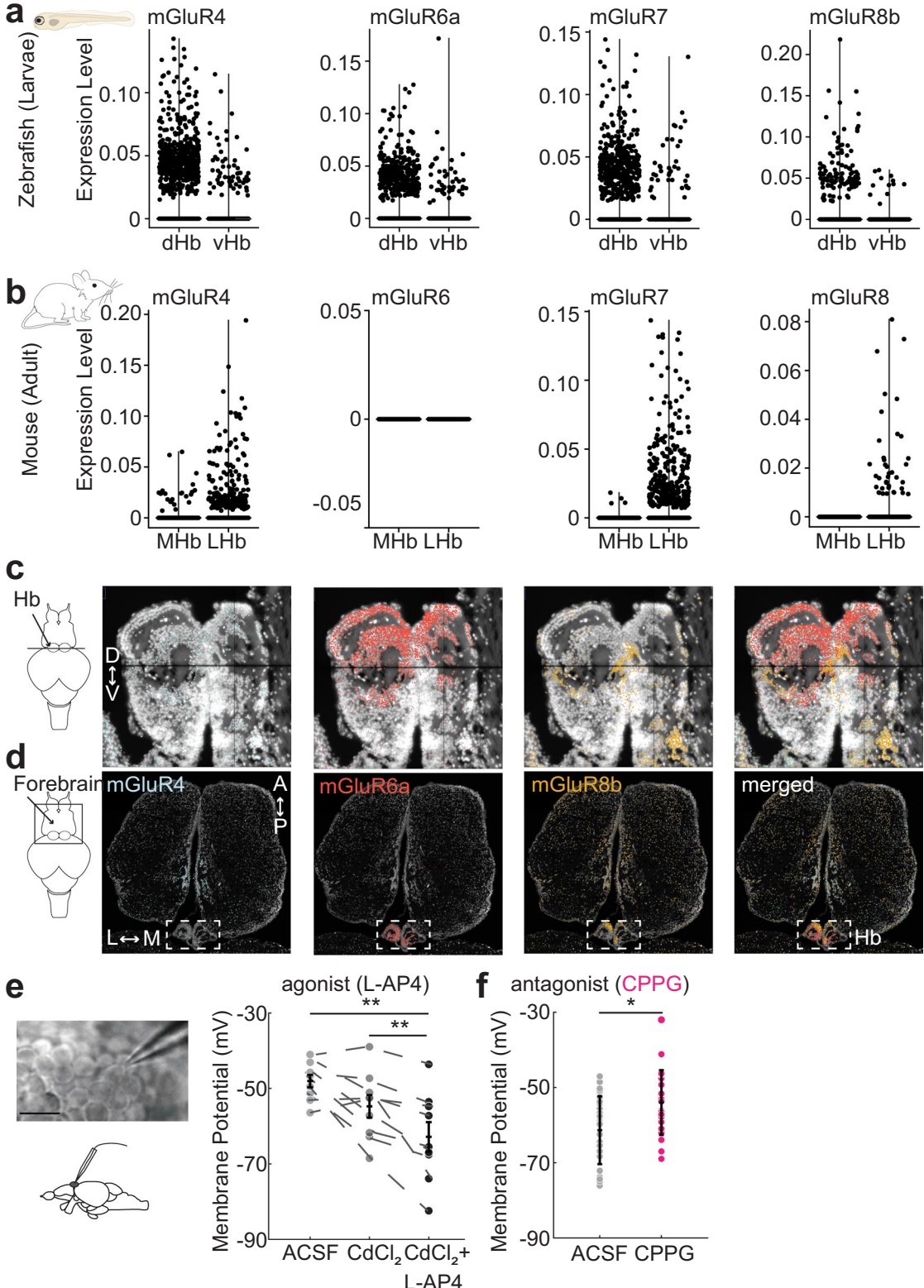

Hb (Fig. 3C). CPPG had no significant effect in the midbrain (Supplementary Fig. 3A, B). In line with these results, group III mGluR agonist L-AP4 (0.1 μM) led to significant reduction of calcium signals, in a significant fraction of Hb and the olfactory bulb (Fig. 3D, Supplementary Fig. 3C).

Our spatial and single cell transcriptomic results revealed that group III mGluR expression is more prominent in dHb as compared to vHb (Fig. 2A, C). To test whether the group III mGluRs are less effective

in controlling vHb, we performed dual channel two-photon calcium imaging in *Tg(elalv3:GCaMP6-nuclear)* zebrafish where a population of vHb neurons was specifically labeled in red by *Tg(dao:GAL4VP16; UAS-E1b:NTR-mCherry)* (Fig. 3E). We identified that the number of DAO-positive vHb neurons responding to the mGluR agonist L-AP4 was significantly smaller than the remaining DAO-negative Hb neurons that are largely in dHb (Fig. 3F, G). Altogether, these results revealed that group III mGluRs are present in zebrafish and rodent Hb, and

**Fig. 2 | Group III mGluRs are expressed in vertebrate habenula and directly regulate neuronal membrane potential. a, b** Scatter plots show the mRNA expression of group III mGluRs in zebrafish dorsal (dHb) and ventral (vHb) habenula (**a**) and mouse medial (MHb) and lateral (LHb) habenula (**b**). **c, d** High-resolution spatial transcriptomics ($n = 2$ fish) for mGluR4 (blue), mGluR6a (red), and mGluR8b (orange) in adult zebrafish habenula (**c**, coronal section), and dorsal forebrain (**d**, horizontal section), along the dorsal-ventral (D dorsal, V ventral), anterior-posterior (A anterior, P posterior) and medial-lateral (M medial, L lateral) axes. Arrow in **c** points to the habenula (Hb) and arrow in (**d**) points to the forebrain. Box in (**d**) with dashed lines indicates habenula. **e** Left: Example image of whole-cell patch clamp recording of a zebrafish habenula neuron. Scale bar represents 10 $\mu$m., Membrane potential measured by whole-cell patch clamp recordings in dHb neurons of juvenile (21dpf) zebrafish brain explants, during control conditions (ACSF,

light grey), with cadmium chloride ($CdCl_2$, 100 $\mu$M, dark-grey) and with $CdCl_2$ (100 $\mu$M) + L-AP4 (10 $\mu$M) (black). Note that group III mGluR agonist L-AP4 decreases the membrane potential of habenula neurons significantly compared to control and $CdCl_2$ conditions. ($n = 9$ fish/neurons, ACSF vs $CdCl_2$ p is n.s., ACSF vs $CdCl_2$ + L-AP4 **$p = 0.0039$, $CdCl_2$ vs $CdCl_2$ + L-AP4 **$p = 0.0039$, two-sided Wilcoxon signed-rank test.). **f** Membrane potential of habenula neurons during control condition (ACSF, grey) and with group III mGluR antagonist CPPG (300$\mu$M, pink). Note that CPPG significantly increases the membrane potential of habenula neurons compared to control conditions. (ACSF $n = 37$ neurons, CPPG $n = 22$ neurons, *$p = 0.0109$, two-sided Wilcoxon rank sum test.). Error bars represent mean +/- SEM. Scattered dots represent individual fish. See also Supplementary Fig. 2. Source data are provided as a Source Data file.

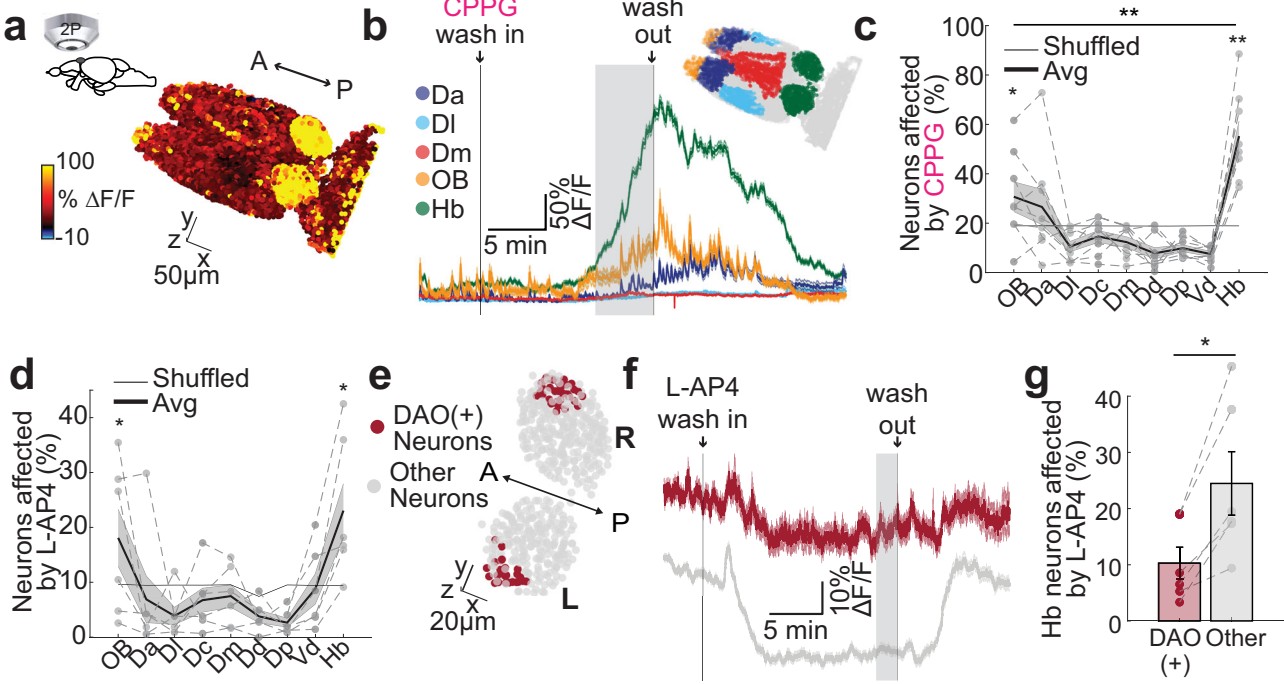

**Fig. 3 | Pharmacological targeting of group III mGluRs can specifically activate and inhibit zebrafish dorsal habenula and the olfactory bulbs. a** 3D reconstructions of calcium signals ($\Delta$F/F) from representative juvenile T*g(elavl3:GCaMP6s-nuclear)* zebrafish forebrain in response to bath application of group III mGluR antagonist CPPG (300 $\mu$M). Warm colors represent stronger activation. **b** Average time courses of calcium signals from anatomically identified forebrain regions (Dorso-anterior (Da): blue, Dorso-lateral (Dl): cyan, Dorso-medial (Dm): red, olfactory bulb (OB): yellow, habenula (Hb): green) in response to CPPG application. Lines indicate the wash in and out, grey area shows the period used to calculate affected neurons in c. Shadow represents +/-SEM. Delineated forebrain regions are color-coded. **c, d** Percentage of neurons in anatomically identified forebrain regions that are affected by CPPG (**c**) or L-AP4 (0.1 $\mu$M) (**d**). Affected means that neuronal calcium signals were 2STD above or below the baseline during 5 min drug period (shaded grey in B or F). Scatters corresponding to individual fish are connected with dashed lines. The thick black line is the mean; shadow presents SEM. Grey line represents the shuffle distribution. Neurons in the olfactory bulb and the habenula are significantly more affected above chance levels (CPPG $n = 8$ fish,

OB *$p = 0.0391$, Hb **$p = 0.0039$; L-AP4 $n = 6$ fish, OB *$p = 0.0469$, Hb *$p = 0.0156$, rest is n.s., one-sided Wilcoxon signed-rank test). Habenula neurons show significantly stronger CPPG responses than olfactory bulb (**$p = 0.0039$, one-sided Wilcoxon signed-rank test). Dc:dorso-central, Dp:dorso-posterior, Vd:ventral-dorsal. **e** Zebrafish habenula expressing T*g(dao:GAL4VP16; UAS-E1b:NTR-mCherry)* and T*g(eval3:GCaMP6-nuclear)*. Dao-positive ventral habenula neurons are indicated in dark red. **f** Average time courses of calcium signals of dao-positive neurons from example habenula in "e" (dark red) and the other habenula neurons (grey). Wash in and out of L-AP4 is indicated by grey lines. The grey area indicated the period for affected cell calculation. Shadow represents SEM. **g** Significantly smaller fraction of dao-positive (DAO(+)) ventral habenula neurons are inhibited by the application of group III mGluR agonist L-AP4, when compared to the rest of habenular neurons ($n = 6$ fish, *$p = 0.0156$, one-sided Wilcoxon signed-rank test). Scatters corresponding to individual fish are connected with dashed lines. L left, R right, A anterior, P posterior. Scale bar represents 50 (**a**) and 20 $\mu$m (**e**). Error bars: mean +/- SEM. See also Supplementary Fig. 3. Source data are provided as a Source Data file.

pharmacological manipulation of group III mGluRs can effectively and specifically alter the membrane potential and activity of Hb neurons, with a stronger preference for the dHb.

## Group III mGluRs regulate sensory responses and selectivity of habenular neurons

Our results showed a significant effect of group III mGluRs in Hb excitability and activity. Next, we asked how group III mGluRs regulate multi-sensory representations in zebrafish Hb. To block group III

mGluRs in vivo, we performed ventricular injection[109,133,134] of CPPG into T*g(elavl3:GCaMP6s)*[109,110,118–120] zebrafish larvae, with group III mGluR expression comparable to juveniles[123]. We next investigated the impact of pharmacological group III mGluR perturbation on sensory representations, by comparing Hb responses of animals injected with 5 mM CPPG to control animals injected with artificial cerebrospinal fluid (ACSF)[130,135]. We choose two different sensory modalities, relatively neutral red-light flashes[105,108,109] and aversive mechanical vibrations[109,136–139]. To focus on Hb specific effects, we avoided the use

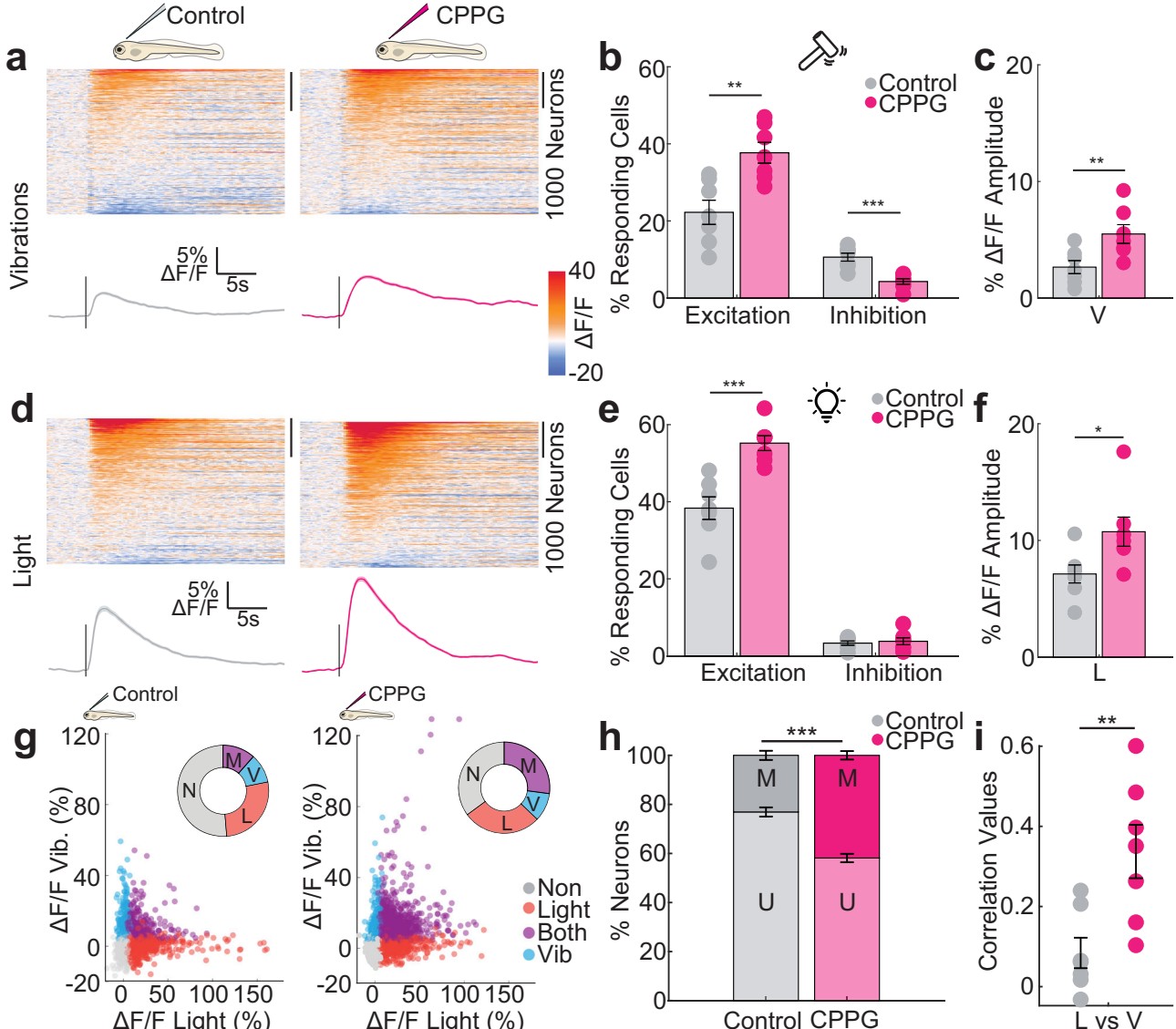

**Fig. 4 | Pharmacological blocking of group III mGluRs amplifies the magnitude and reduces the selectivity of sensory responses of habenular neurons.** Heatmaps represent the time courses of habenular calcium signals (ΔF/F) in responses to vibrations (**a**) or light (**d**) recorded by two-photon calcium imaging in *Tg(e-lavl3:GCaMP6s)* larval zebrafish. Left: control-injected (*n* = 3732 neurons, 7 fish) or right: 5 mM CPPG-injected (*n* = 4024 neurons, 7 fish). Warm colors indicate excitation, cold colors represent inhibition. Average traces of all habenula neurons are below each heatmap. Stimulus onset is indicated by a line. Shadow represents +/-SEM. Percentage of excited (2 STD above baseline) or inhibited (1 STD below baseline) habenula neurons for control (grey) or CPPG-injected (pink) fish in response to mechanical vibrations (**b**) or light (**e**) stimulation. (Control *n* = 7 fish, CPPG *n* = 7 fish, vibration: excitation **$p$ = 0.0020, inhibition ***$p$ = 0.0003; light: excitation: ***$p$ = 0.0003, inhibition: p is n.s., one-sided Wilcoxon rank sum test). **c,f** Average ΔF/F Amplitude (%) during the response period of all habenula neurons in

each fish. (Control *n* = 7 fish, CPPG *n* = 7 fish, vibration: **$p$ = 0.0087; light: *$p$ = 0.0189, one-sided Wilcoxon rank sum test). **g** Responses of individual habenula neurons to mechanical vibration (blue), light (red) or both (magenta) for control and CPPG injected fish. The donut chart represents the ratio of habenula neurons and their response type (2 STD above baseline levels). N: non-responding, V: only vibrations, L: only light, M: both vibrations and light. **h** Percentage of unimodal (U) habenula neurons that respond exclusively to either light (L) or vibrations (V) versus multimodal (M) neurons responding to both light and vibrations. (Control *n* = 7 fish, CPPG *n* = 7 fish, ***$p$ = 0.0003, one-sided Wilcoxon rank sum test). **i** Pearson's correlation of multi-neuronal response vectors in the habenula for mechanical vibrations (V) and light (L) (Control *n* = 7 fish, CPPG *n* = 7 fish, **$p$ = 0.0035, one-sided Wilcoxon rank sum test). Error bars represent mean +/- SEM. Scattered dots represent individual fish. See also Supplementary Figs. 4, 5, 6 and 7. Source data are provided as a Source Data file.

of odors due to the expression and impact of group III mGluRs in the OB (Supplementary Fig. 2A, Fig. 3C-D). We observed that a significantly larger fraction of Hb neurons exhibit excitatory calcium responses to aversive mechanical vibrations in CPPG-injected animals (Fig. 4A, B). Such increase in excitatory vibration responses in CPPG-injected animals are independent of response thresholds (Supplementary Fig. 4). We also saw a significant reduction of Hb neurons with inhibitory responses to vibrations. This led to an overall increase in average Hb response to vibrations (Fig. 4C). Similarly, light responses in Hb also showed a significant increase in CPPG-injected animals (Fig. 4D–F,

Supplementary Fig, 4). Next, we asked whether this significant increase in Hb responses also lead to a reduced selectivity of Hb neurons for these distinct sensory modalities. We observed a significant increase in the number of Hb neurons responding to both sensory modalities (multimodal) in CPPG-injected animals and a decrease in unimodal (selective) neurons (Fig. 4G, H, Supplementary Fig. 4C–F-I). In fact, multi-neuronal sensory representations of vibration and light stimuli were significantly more correlated (Fig. 4I), exhibit higher cosine similarity (Supplementary Fig. 4J), and hence more similar in CPPG-injected animals. The control injections did not alter sensory

responses in Hb (Supplementary Fig. 5). CPPG injections did not alter midbrain sensory responses (Supplementary Fig, 6). The impact of CPPG-injections on habenular sensory computations was more prominent on dorsal regions of Hb, when compared to ventral regions (Supplementary Fig. 7). Taken together, blocking group III mGluRs increases sensory-evoked excitation and decreases the selectivity of Hb neurons to vibrations and light stimulation.

### Inhibitory interactions among glutamatergic dHb neurons are mediated by group III mGluRs

Zebrafish Hb neurons were shown to exhibit spatio-temporally structured spontaneous (or resting-state) activity[107,109,110]. Similarly, we observed that positively correlated Hb neurons are closer to each other as compared to negatively correlated neurons in *Tg(elavl3:GCaMP6s)* zebrafish larvae (Fig. 5A, B). This can also be observed by a decrease in average pairwise Pearson's correlations as a function of distance between Hb neurons (Fig. 5C, grey). Blocking group III mGluRs by ventricular CPPG injection resulted in an increase in pairwise correlation of spontaneous calcium signals (Fig. 5C, pink), suggesting that group III mGluRs regulate the spatial features of spontaneous Hb activity, likely due to disinhibition of Hb ensembles.

The spatial organization of spontaneous activity in zebrafish Hb is likely due to spatially restricted inputs to Hb, as shown for olfactory[86,107], visual[84,85,109] and hypothalamic[74] innervations. Another possible contributor to such functional interactions (correlations and anticorrelations) of Hb neurons are direct or indirect connections between them. To test, whether Hb neurons can communicate with each other, we performed patch clamp recordings to stimulate individual dHb neurons, while performing calcium imaging of surrounding Hb neurons in brain explants of *Tg(elavl3:GCaMP6s-nuclear)* juvenile zebrafish (Fig. 5D). We observed a rise of calcium signals in the microstimulated neuron (Fig. 5D, brown) followed by an exponential decay. Upon micro-stimulation of single Hb neurons, we also observed small but significant excitation (Fig. 5D, orange) and inhibition (Fig. 5D, blue) in Hb neurons surrounding the stimulated neuron (Fig. 5D, E). We observed a larger fraction of Hb neurons were inhibited (decreased calcium signals) than excited (increased calcium signals) (Fig. 5E). Excited Hb neurons were significantly closer to micro-stimulated neurons than inhibited ones (Fig. 5F). Blocking group III mGluRs by bath application of 300 µM CPPG, did not affect the fraction of excited Hb neurons, but reduced the fraction inhibited neurons upon micro-stimulation of individual Hb neurons (Fig. 5G, H). These results suggest that mGluRs may regulate inhibitory/competitive interactions between Hb neurons.

Axon terminals of Hb neurons were shown to exhibit competitive interactions via presynaptic GABA$_B$ receptors through Hb-interpeduncular connections[116]. Our results with CPPG perturbation of inhibitory interactions between Hb neurons (Fig. 5D–H) revealed that group III mGluRs might play a complementary role. We asked whether primarily glutamatergic dHb neurons[109,111] exhibit synaptic vesicles within their processes in densely arborized core of Hb. To answer this, we used the *Tg(narp:GAL4VP16;UAS: Synaptophysin-GFP-T2A-tdTomato-CAAX)*[63,122] zebrafish, labeling neural membrane in red and synaptic vesicles in green. The dHb mainly projects to the interpeduncular nucleus (IPN)[59,108,140,141]. As expected, we observed that the majority of bright *Synaptophysin-GFP* (SypGFP) expression was at the IPN (Supplementary Fig. 8A). Complementary to this, we also observed substantial *Synaptophysin-GFP* expression within the core region of dHb (Fig. 5I-K), innervated by dHb neuron dendrites, suggesting that glutamatergic dHb neuron dendrites express presynaptic markers within Hb.

Altogether, our findings revealed that glutamatergic dHb neurons can communicate with each other primarily through inhibition, which is mediated, at least in part, through group III mGluRs.

### Multi-sensory stimulus competition in the habenular circuits is partially dependent on group III mGluR

We observed prominent inhibitory interactions within Hb neurons, and an important role of group III mGluRs in such interactions and sensory representations in Hb. We hypothesized that such inhibition across distant Hb neurons might be important for competition between sensory representations, when different sensory cues are presented. To test this hypothesis, we first presented light and vibrational stimuli separately and later simultaneously and measured the sensory responses in vivo. Comparing simultaneous or combinatorial delivery of different stimuli to individual light and vibrational responses revealed three different kinds of neurons, super-additive (Fig. 6A, orange), depressed (Fig. 6A, blue), and neither super-additive nor depressed, which we termed sub-additive. Calculating an interactions index[142,143] for these categories (Fig. 6B), revealed that around 60% of Hb neurons showed depressed responses to simultaneous delivery of light and vibrations, when compared to responses to individual stimuli (Fig. 6C). Less than 20% of Hb neurons demonstrated super-additive interactions. Hence, most interactions between light and vibrational responses in Hb are depressive (or inhibitory), when both stimuli are delivered simultaneously. Next, we asked to what extent group III mGluR dependent interactions between distant Hb neurons play a role in such depressive interactions between sensory modalities. We observed that ventricular injections of 5 mM CPPG, significantly reduced the fraction of depressive interactions between light and vibrational stimuli, while increasing the amount of sub-additivity (Fig. 6D, E). These results showed that competition is the main mode of interaction between sensory representations in Hb and inhibition via group III mGluRs plays a prominent role in such competition.

### Habenula specific perturbations of group III mGluRs alter sensory responses in the dorsal habenula and amplifies defensive behaviors

Previous research has shown that Hb is involved in various defensive and adaptive behaviors[57–64,66,92,101–103]. To further reveal the role of group III mGluR mediated inhibition in Hb sensory-computations and sensory-evoked defensive behaviors, we performed a set of calcium imaging and behavioral experiments in knockout fish lacking mGluR6a protein[144], the primary group III mGluR expressed in Hb[145] (Fig. 2D, red). We used 3-weeks-old juvenile zebrafish that can generate cognitively demanding behaviors[62,67,146–150].

We first investigated to what extent mGluR6a knockout, with no known physiological phenotype[144], recapitulated the pharmacological alterations that we observed in the zebrafish injected with group III antagonist CPPG (Fig. 4). Since mGluR6a transcripts are primarily present on dorso-medial Hb (dmHb) (Supplementary Fig. 9A), we focused our functional analysis on the Hb neurons that are in the most dorso-medial regions (Supp. Fig. 9B). Two-photon calcium imaging of juvenile *Tg(elavl3:GCaMP6s)* zebrafish revealed that sensory responses in mGluR6a mutants are elevated (Supplementary Fig. 9C–H), and less stimulus-specific (Supplementary Fig. 9I–K). These results are very similar to the changes in sensory representations we observed upon blocking group III mGluRs via ventricular CPPG injections (Fig. 4).

Next, we asked whether the behavior of mGluR6a mutant zebrafish is altered in well-established assays for defensive and adaptive behaviors. First, we tested mGluR6a mutants in a novel tank test[151,152], where zebrafish exhibit a defensive response by diving at the bottom of the new tank before gradually exploring the tank at various depths[153–158]. We observed that while control fish initiate such exploratory behavior and swim up, already within first 2 min of exposure to new tank, heterozygous and homozygous mGluR6a mutants remained significantly closer to the bottom of the tank (Fig. 7A, B). Consistently, CPPG-injected juvenile zebrafish viewed from the top, exhibit significantly smaller adaptation of their swim speed, like novel-tank response, when introduced in a new arena (Supplementary

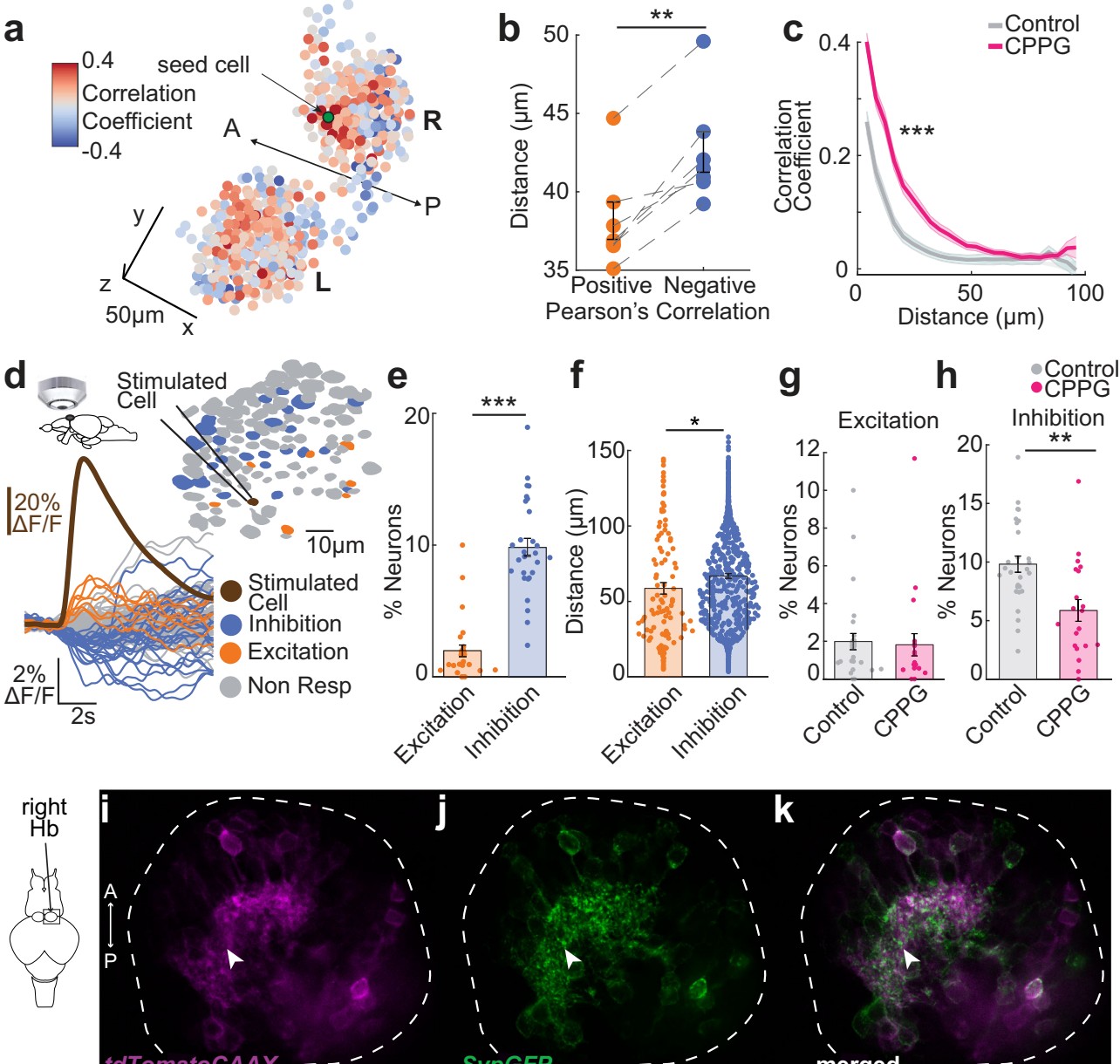

**Fig. 5 | Group III mGluRs play an important role in coordinating spontaneous habenular activity and mediating inhibitory interactions between habenular neurons. a** Pearson's correlations of spontaneous activity between Hb neurons and a seed neuron (green) in *Tg(elavl3:GCaMP6s)* larval zebrafish. Red means high and blue low correlations. Scale bar represents 20 μm. L left, R right, A anterior, P posterior. **b** Average distance between Hb neuron pairs that are significantly positively or negatively correlated. Positive correlated neurons are significantly closer than negative ones. (*n* = 7 fish, **p = 0.0035, one sided Wilcoxon signed-rank test). **c** Pairwise Pearson's spontaneous activity correlation of habenula neurons as a function of distance (μm) in control (grey) versus 5 mM CPPG-injected fish (pink). CPPG-injected animals exhibit stronger correlations over longer distances. Shadow represents +/-SEM. (Control: *n* = 14 hemispheres from 7 fish, CPPG-injected: *n* = 14 Hb hemispheres from 7 fish, ANOVA-n displayed significance over distances ***p = 1.8 × 10⁻¹¹ and over treatment groups ***p = 0.0004). **d** Calcium imaging of *Tg(eval3:GCaMP6-nuclear)* juvenile zebrafish brain explant while simultaneously stimulating a single cell via whole-cell recording, brown: stimulated neuron, orange: excited, blue: inhibited, grey: non-responding. Example calcium traces are shown on the left (ΔF/F). **e** Percentage of habenula neurons excited or inhibited upon single habenular neuron micro-stimulation. Significantly more cells are inhibited than excited (*n* = 28 stimulated individual neurons, ***p = 0.00007, two-sided Wilcoxon signed-rank test. **f** Distance (μm) of the responding neuron

(excited, orange or inhibited: blue) to the micro-stimulated neuron. Inhibited neurons are significantly more distant to the stimulated neurons than the excited ones (*n* = 104 excited neurons and *n* = 558 inhibited neurons after 28 stimulated neurons, *p = 0.0291, two-sided Wilcoxon signed-rank test)(**g**, **h**) Percentage of habenula neurons increasing (Excitation, **g**) or decreasing (Inhibition, **h**) their fluorescence upon single cell stimulation during control conditions or during bath application of 300 μM CPPG. Significantly less cells are inhibited when 300 μM CPPG is applied, but no difference for the fraction of excited neurons. (ACSF *n* = 28 stimulated neurons; CPPG *n* = 19 stimulated neurons, Excitation p is n.s; Inhibition **p = 0.007 two-sided Wilcoxon rank sum test). Note that the control data is same as in (**e**). **i**–**k** Confocal microscopy images of tissue-cleared *Tg(narp:GAL4VP16;UAS:Synaptophysin−GFP-T2A-tdTomato-CAAX)* juvenile zebrafish (*n* = 5 fish). Colors represent *tdTomatoCAAX* (magenta) and *Synaptophysin−GFP* (Syp-GFP, green). Dorsal habenula neurons expressing *tdTomatoCAAX* (**i**), *Synaptophysin−GFP* (**j**) and merged (**k**). White line delineate the habenula. Scale bar represents 10 μm. White arrow points at *Synaptophysin−GFP* on the dendritic processes of *narp* labelled dorsal habenula neurons. A anterior, P posterior. See also Supplementary Fig. 8. Error bars represent mean +/- SEM. Scattered dots represent individual fish (**b**) or neurons (**e**–**h**). Source data are provided as a Source Data file.

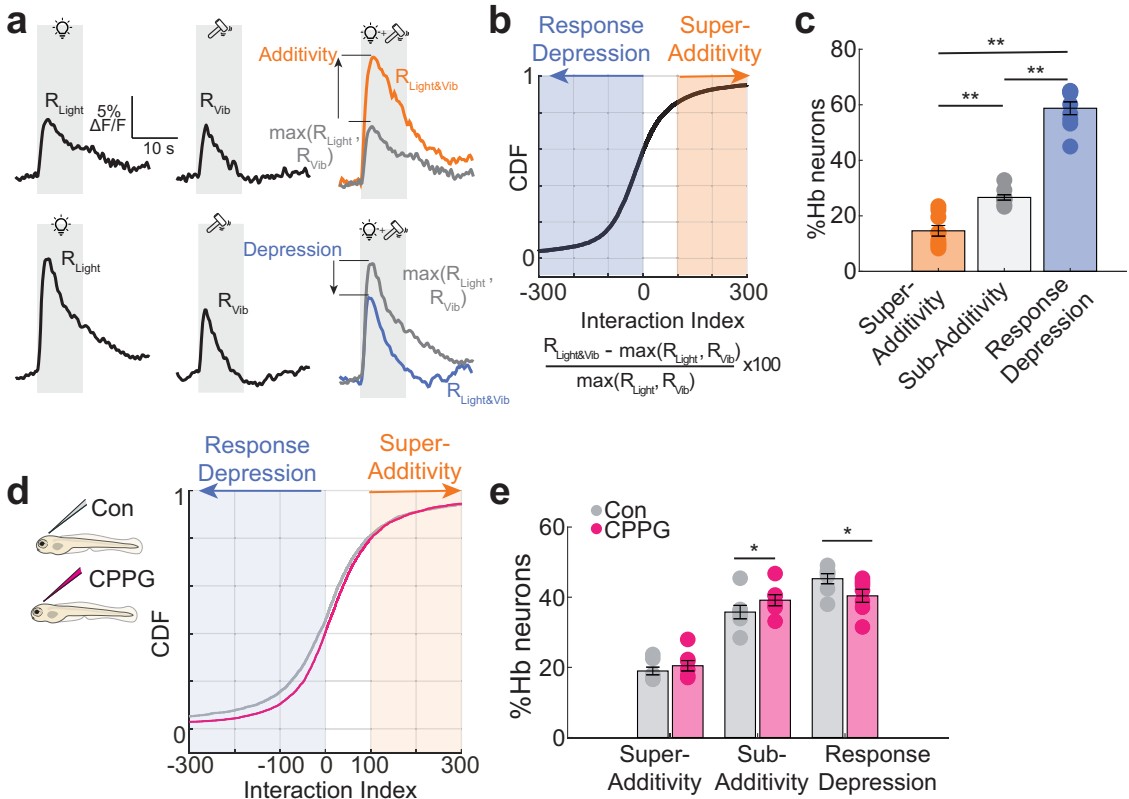

**Fig. 6 | The role of group III mGluRs in multi-sensory stimulus competition in zebrafish habenula. a** Example traces of habenula neurons responding to light, vibration and light and mechanical vibrations simultaneously. Upper traces show an example of additivity (orange), where combined light+vibration response ($R_{Light\&Vib}$) is larger than the largest individual light and vibration responses ($R_{Light}$ and $R_{Vib}$ black). Bottom traces show an example of depression (blue) where combined light+vibration response is smaller than the largest of individual light or vibration responses (black). In grey, the maximum response (either light or vibration, $\max(R_{Light}, R_{Vib})$) is shown. Grey traces near colored traces show the comparison of combined response to the largest of either light or vibration responses. **b** Cumulative distribution function of the interaction index for all habenula neuron in each fish ($n = 5242$ neurons in 9 fish). Values below 0 (blue shadow) are classified as response depression, values above 100 (orange shadow) are classified as super-additivity. Values between 0 and 100 are classified as sub-additivity. Data points above 300 and below −300 are not shown. **c** Percentage of neurons in categories of super-additivity, sub-additivity and response depression after calculation of the interactive index. There are significantly more neurons showing response depression than super-additivity or sub-additivity ($n = 9$ fish, Super-Additivity vs Sub-Additivity **$p = 0.0039$, Sub-Additivity vs Response Depression **$p = 0.0020$, Super-Additivity vs Response Depression $p = 0.0020$, one-sided Wilcoxon signed-rank test). **d** Interactive index of habenula neurons from control-injected (grey, $n = 3732$ neurons in 7 fish) and 5 mM CPPG-injected fish (pink, $n = 4024$ neurons in $n = 7$ fish). Data points above 300 and below −300 are not shown. **e** Percentage of neurons falling into the categories after calculation of the interaction index. Significantly less neurons show response depression in CPPG-injected fish as well as more cells falling into the sub-additivity category. (Control $n = 7$ fish, CPPG $n = 7$ fish, Super-Additivity p is n.s., Sub-Additivity *$p = 0.0487$, Response Depression *$p = 0.0189$, one-sided Wilcoxon rank sum test). Error bars represent mean +/- SEM. Scattered dots represent individual fish. Source data are provided as a Source Data file.

Fig. 10A, B). Subsequently, we tested the behavioral response of zebrafish to mechanical vibrations, which are known to generate a reflex-like startle response[109,136–139]. We observed that upon mechanical vibrations juvenile zebrafish generate an immediate and defensive[159] bottom diving response followed by sustained bottom swimming that outlasts the short (4 s) vibrational stimuli for up to 30 s (Fig. 7C). We observed that mGluR6a mutants remained significantly closer to the bottom of the tank during this sustained behavior and are slower in adapting their swim depth long after the vibration stimuli. Similarly, CPPG-injected juvenile zebrafish viewed from the top exhibit a reflex-like initial increase in swim speed, but then swim significantly slower after exposure to mechanical vibrations (Supp. Fig. 10C, D). Finally, we used a light-dark transition assay that is typically used to test anxiety-like behaviors in zebrafish[160–163]. In this test, turning off the light generated an initial startle reflex-like response followed by sustained swim with increased speeds in a horizontal arena (Fig. 7E). We observed that both mGluR6a mutants (Fig. 7E, F) and CPPG-injected juvenile zebrafish (Fig. 7G, H) maintained significantly higher sustained swim speeds during light-dark transitions. In summary, we observed amplified defensive behaviors evoked by environmental changes and sensory stimulation in juvenile zebrafish with perturbed group III mGluRs. Altogether our findings revealed an important role for group III mGluR mediated inhibition in Hb in sensory computations, neural connectivity, and animal behavior.

## Discussion

In this study we investigated the role of group III mGluR-dependent inhibition in brain function and animal behavior. We showed that glutamate-dependent inhibition can mediate several functional features that are usually attributed to GABA, in the Hb, a brain region that largely lacks GABAergic neurons[109,111,113,117]. By pharmacological and genetic perturbation of group III mGluRs, we showed that Hb neurons exhibit larger and less specific responses to different sensory modalities. We revealed that glutamatergic Hb neurons can interact with each other, largely through inhibition, which is sensitive to group III mGluR blockers. In line with this, we also observed that the default interaction between distinct sensory representations in Hb is mostly based on suppression and depends on group III mGluRs. Genetic perturbation of Hb specific mGluR6a gene, and pharmacological perturbation of group III mGluRs led to amplified defensive responses to

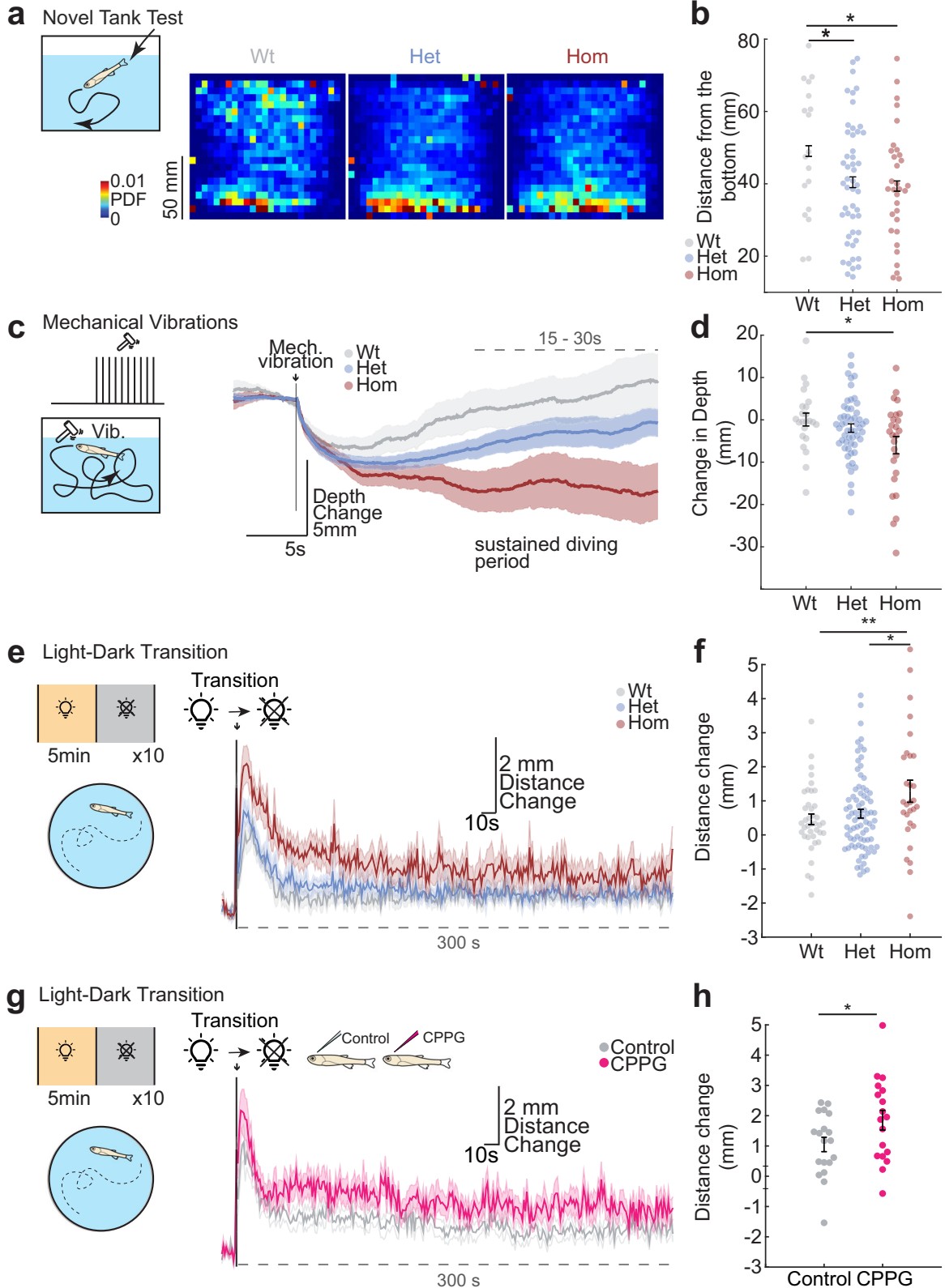

sensory stimuli and environmental changes. All these results indicate that group III mGluRs, a rather understudied class of mGluRs in the central nervous system, can mediate features of Hb connectivity and computations that are typically attributed to GABAergic inhibition in other parts of vertebrate forebrain.

The Hb receives inputs from various forebrain regions and sensory systems[65,66,74–80,84–86,107–110]. The LHb/vHb receives both

glutamatergic[66,164–167] and GABAergic[66,78,164,168] inputs. To date, less is known about the GABAergic inputs into MHb/dHb as most reports have studied primarily the glutamatergic system[169]. Importantly, sensory inputs to the dHb in zebrafish are primarily glutamatergic[84–86,122] and MHb/dHb neurons are glutamatergic as well[109,111]. Nevertheless, sensory-evoked inhibition is present in dHb[107–109]. Hence, the GABA-poor, dHb appears to have evolved parallel inhibitory pathways. A

**Fig. 7 | Genetic perturbation of mGluR6a enhances sustained defensive behaviors in juvenile zebrafish. a** Novel vertical tank diving test of free-swimming in wildtype (Wt, grey), heterozygous (Het, blue), homozygous (Hom, red) mGluR6a mutant zebrafish. Average probability density (PDF) of the fish position, during the first 2 minute in the novel tank. Warmer colors mean higher probability. **b** Distance from the bottom (mm) during the first 2 min. Note that Het and Hom are significantly closer to the bottom when compared to Wt fish (Wt $n = 18$ fish, Het $n = 50$ fish, Hom $n = 32$ fish, Wt vs Het $*p = 0.0405$, Wt vs Hom $*p = 0.0337$, Het vs Hom is n.s., one-sided Wilcoxon rank sum test). **c** Mechanical vibrations are delivered to free-swimming juvenile zebrafish (vertical tank). Average diving depths evoked by mechanical vibrations. Dashed line indicates the sustained period used for statistical comparisons in (**d**). **d** Sustained diving depth following mechanical vibrations is shown for the three groups. Homozygous mGluR6a mutants dive significantly deeper (Wt $n = 23$ fish, Het $n = 54$ fish, Hom $n = 26$ fish. Wt vs Het p is n.s., Wt vs Hom $*p = 0.0231$, Het vs Hom is n.s., one-sided Wilcoxon rank sum test). **e, g** Sustained increase in free swimming speed evoked by Light-Dark transition (horizontal tank). Average swimming distances (mm) are plotted for mGluR6a mutant Hom: red, Het: blue, Wt: grey in (**e**) and for 5 mM CPPG- (pink) and control-(grey) injected juvenile fish. Arrow indicate the light to dark transition. Dashed lines indicate the sustained increase in swimming rate evoked by light-dark transition used for statistics in (**f**) and (**h**). **f** Homozygous mGluR6a mutants swim significantly more (mm) upon light and dark transition (Wt $n = 39$ fish, Het $n = 80$ fish, Hom $n = 29$ fish Wt vs Het p is n.s., Wt vs Hom $**p = 0.0068$, Het vs Hom $*p = 0.0178$., one-sided Wilcoxon rank sum test). **h** CPPG-injected fish swim significantly more (mm) upon light and dark transition (Control $n = 19$ fish, CPPG $n = 18$ fish, $*p = 0.0288$, one-sided Wilcoxon rank sum test). Shadow indicates +/-SEM. Error bars represent mean +/-SEM. Scattered dots represent individual fish. See also Supplementary Fig. 9 for imaging data of the mGluR6a mutant and Supplementary Fig. 10 for more behavior of the CPPG-injected fish. Source data are provided as a Source Data file.

recent study shows that dHb terminals innervating interpeduncular nucleus exhibit presynaptic inhibition through the feedback from GABAergic interpeduncular neurons, leading to competition between cholinergic and non-cholinergic dHb neurons[116]. We now showed a parallel and perhaps complementary inhibition system for dHb neurons that relies on glutamate. mGluR-dependent inhibition of dHb neurons may be triggered by glutamate release through massive forebrain sensory-limbic glutamatergic innervations to Hb, as well as by the glutamatergic Hb neurons themselves. In fact, our results in Fig. 5, showing *synaptophysin-GFP* expression in the core dendritic region of dHb, together with single-neuron stimulation experiments that inhibit surrounding Hb neurons, raise the possibility of dendro-dendritic synapses between dHb neurons. This idea of dendro-dendritic synapses should be tested in future high-resolution electron microscope reconstructions of Hb. It is possible that both mechanisms of glutamate release (by Hb inputs and by Hb-Hb connections) are at play, or that one of them dominate in triggering mGluR-dependent inhibition. Group III mGluRs are thought to be present both in the synapse and extrasynaptically[49,170], and mGluR-dependent inhibition relies on multiple cellular cascades[49–51]. These factors contribute to the complex nature of mGluR-dependent inhibition. Further pharmacological dissection of group III signaling cascades in the vertebrate Hb would be interesting for future studies. Regardless of the source of glutamate, our results demonstrate that mGluR-dependent inhibition plays a crucial role in shaping sensory computations in the Hb and regulating animal behavior.

We showed that suppression is the default integration mode of Hb neurons for distinct sensory inputs, which resembles lateral inhibition in sensory systems[9,11,171–175]. Such cross-modal inhibition is in line with the proposed function of Hb to act as a switchboard[57,76,107,109,146]. In fact, our results suggest that suppression across different modalities as well as Hb ensembles might be the primary substrate for such a switchboard function of Hb. This cross-modal inhibition in Hb is at least partly mediated by group III mGluRs receptors in GABA-poor dHb/MHb, as well as by multi-synaptic inhibition through Hb-interpeduncular nucleus (IPN) connections onto presynaptic terminals of Hb neurons[116]. Both dHb/MHb and vHb/LHb are instrumental for adaptive behaviors[57–64,66,68,92,101–103] and mood disorders[71–73]. While different group III mGluRs were shown to be found across different zebrafish brain regions[123], the intersection of expression patterns for multiple group III mGluRs in dHb is remarkable. This likely explain why we have seen strongest effects of pharmacological group III mGluR perturbations in dHb neurons and not in other forebrain and midbrain regions, nor in ventral zones of Hb. We observed only minimal impact of group III mGluRs in zebrafish vHb, which was shown to receive GABAergic terminals that likely mediate most observed inhibition in vHb[176]. However, given prominent expression of group III mGluRs in rodent LHb and MHb such inhibition can still play unknown roles in mammalian Hb. Evolutionary conservation between zebrafish and mouse Hb[117] further justifies future studies on the function of group III mGluRs in mammalian Hb. Coincidentally, few studies showed noticeable effects on defensive and anxiety-like behaviors in models of mood disorders, upon perturbation of group III mGluRs, mGluR4 and mGluR8 in various mammalian forebrain regions[52,177–184]. Given relatively specific responses of zebrafish Hb to group III mGluR pharmacology, we proposed that group III mGluRs can be a potential drug target for interfering with Hb function and mood disorders.

In our study, we observed several neural circuit features such as coordinating ensembles[114,185,186], cross modality suppression[24,142,187], sensory selectivity and gain[10,12,14,18,20,23] that are attributed to GABAergic circuits across the vertebrate forebrain, are at least in part mediated by group III mGluRs in dHb. Yet, group III mGluRs are vastly expressed throughout mammalian brain[50,52,53,188,189]. While few studies investigated the function of group III mGluRs in retina[50,190,191], we know very little about the role of mGluR-dependent inhibition in brain function and animal behavior. An interplay between group III mGluRs and GABAergic interneurons has been seen in a recent study[192], which suggests glutamatergic and GABAergic inhibition may complement each other. Recently available molecular atlas of mammalian brain[54,193,194] suggests broad expression of group III mGluRs such as mGluR7 and mGluR8 in various cortical areas, from entorhinal to sensory/motor and associative cortices. These findings raise an important question about the function of group III mGluRs for neural computations across the mammalian cortex. Interestingly, group III mGluRs were suggested as drug targets for various human disorders like neurodegenerative diseases[50,195], anxiety[177] and absence epilepsy[196]. Our results in Hb and these previous studies across the brain highlight the potential of studying group III mGluRs function from the perspective of brain physiology and pathophysiology. Future studies will reveal what other functions glutamate driven inhibition may play in vertebrate brain.

## Methods

### Zebrafish maintenance and strains

The animal facility and fish maintenance were approved by the NFSA (Norwegian Food Safety Authority). Zebrafish, *Danio rerio*, were kept in 3.5 L tanks in a Techniplast ZebTec Multilinking System with constant conditions maintained (28.5 C, pH 7.2, 700 uSiemens, 14:10 h light/dark cycle). The fish were fed with dry food (SDS100 up to 14 dpf and SDS400 for adult animals, Tecnilab BMI, the Netherlands) twice a day and *Artemia nauplii* (Grade 0, Platinum Label, Argent Laboratories, Redmond, USA) once a day. Larvae from fertilization to 3 dpf were kept in a Petri dish with egg water (1.2 g marine salt in 20 L RO water, 1:10000 0.01% methylene blue) and in artificial fish water (AFW: 1.2 g marine salt in 20 L RO water) between 3 and 5 dpf. Larval (5dpf) and juveniles (3 to 4-week-old) zebrafish were used for the experiments and analyzed irrespective of gender. The following fish lines were used for experiments: *Tg(elavl3:GCaMP6s)*[120], Tg(*elavl3:GCaMP6s-nuclear)*[120], Tg(*dao:GAL4VP16;*

*UAS-E1b:NTR-mCherry)*[58],*Tg(narp:GAL4VP16;UAS:Syp−GFP-T2A-tdTomato-CAAX)*[63,122] mGluR6a mutant was created using a CRISPR/Cas9 mediated knockout.

## Inclusion and ethics statement

Our team includes researchers from diverse origins and backgrounds. One or more of the authors of this paper self-identifies as an under-represented minority in science.

All experimental procedures that were performed on zebrafish larvae and juveniles were in accordance with the Directive 2010/63/EU of the European Parliament and the Council of the European Union. They were approved by the Norwegian Food Safety Authorities. Animals from both sexes were used in this study.

## Two-photon calcium imaging

For calcium imaging two photon microscopes (Scientifica) in a combination with a 16x water immersion objective (Nikon, NA 0.8, LWD 3.0) and a 20x water immersion objective (Zeiss 7 MP, W Plan-Apochromat NA 1.0) were used. A mode-locked Ti:Saphire laser (Mai-Tai Spectra Physics) tuned to 920 nm was used for excitation. In addition, recordings were either performed single plane or volumetric (8 planes using a Piezo (Physik Instrumente (PI))). Volumetric recordings had an acquisition rate of 2.3–3.4 Hz per plane and an average images size of $1536 \times 600$ pixel.

Fish for in vivo imaging were embedded in low melting point agarose (2–2.5% LMP, Fisher Scientific) inside a recording chamber (Fluorodish, World Precision Instruments) for head-restraining. After 20 min the LMP agarose solidified and the section covering the mouth and nose was carefully removed for juvenile zebrafish but left intact for larval ones. The embedded fish was then placed under the microscope and in case of the juvenile zebrafish constantly perfused with bubbled AFW. Before any sensory stimulations ongoing activity was always recorded for 10 min. All our sensory stimulation parameters are selected based on our earlier studies[105,108,109]. A red LED (LZ1-00R105, LedEngin; 630-nm peak wavelength, for more information see data sheet: https://eu.mouser.com/datasheet/2/588/lz1_00r105-2891337.pdf) was used for light stimulations and placed it in the front of the recording chamber. The duration of the light stimulus was a flash of ambient light for 2 ms with a measured intensity of 0.318 mW at 635 nm, at the location of fish during these experiments. Mechanical vibrations were delivered via solenoid tapper (SparkFunElectronics, ROB-10391), via 50 ms application of 6 V. The properties of the vibration stimuli in dB and Hz are presented in Supplementary Fig. 3D. For co-stimulations of the two sensory stimuli, the onsets were synchronized. Light, vibration or co-stimulation of light and vibration were repeated eight times with a 60 s interstimulus interval, and 4 min were between the different sets of stimulus conditions.

For odor stimulation we prepared an amino acid mixture (Alanine, Phenylalanine, Methionine, Histidine, Cysteine, Arginine, Glutamic acid at $10^{-4}$ M), and ammonium chloride ($10^{-4}$ M). For the Supplementary Fig. 1, we also used two other sets of odors: bile-acid mixture (taurocholic acid, taurodeoxycholic acid at $5 \times 10^{-4}$ M) and food odor (1 g/50 ml dilution). All odorants were purchased from Sigma Aldrich. Food odor was prepared with commercially available fish food. For at least one hour, 1 g of food particles was incubated in 50 ml of fish water, then filtered through filter paper, and lastly diluted to 1:50. The odor was delivered via stimulation tube that was placed in front of the nose of the fish. The stimulation lasted 30 s and was performed using a HPLC injection valve (Valco Instruments) controlled with Arduino Due. To determine the precise onset of odor delivery a trial with fluorescein ($10^{-4}$ M AFW) was performed before each experiment.

For the whole brain explant imaging, the dissection of the juvenile zebrafish explant was described previously[110]. In short, animals were anesthetized in ice-cold AFW and euthanized by decapitation in oxygenated (95% $O_2$/5% $CO_2$) ACSF. The ACSF was composed of the following chemicals diluted in reverse osmosis-purified water: 131 mM

NaCl, 2 mM KCl, 1.23 mM KH2PO4, 2 mM MgSO47H2O, 10 mM glucose, 2.5 mM CaCl2, and 20 mM NaHCO3 as previously explained[130]. Both, the dissection, and experiment with the brain explant was carried out in bubbled ACSF. After removing the jaw and eyes, the muscle tissue, gills, and fat were cleaned of to ensure proper oxygen diffusion of the explant. Then, bones, skin tissue and dura mater covering the forebrain and/or Hb were removed from the dorsal side. The brain explant was then mounted using tungsten pins to a small petri dish coated with Sylgard (World Precision Instruments) and perfused in constant flow bubbled ACSF. The brain explant is consistently perfused with artificial cerebrospinal fluid (ACSF) that is bubbled with carbogen (95% $O_2$ and 5% $CO_2$) throughout the entire duration of the experiment.

## Electrophysiological recordings

Patch clamp recordings were conducted in a juvenile brain explant. For intracellular recordings of Hb neurons, borosilicate glass capillaries of 9–15 MOhms were filled with intracellular solution which contained (in mM): 130 K-gluconate, 10 Na-gluconate, 10 HEPES, 10 $Na^{2+}$-Phospho-Creatine, 4 NaCl, 4 ATP-Mg and 0.3 $Na^{3+}$-GTP[130]. Electrical signals were recorded by MultiClamp 700B amplifier at sampling rate of 10 KHz. All recordings and data analyses were performed using custom codes written in MATLAB. To perturbed group III mGluRs the agonist, L-AP4 (Tocris) with a concentration of 10 $\mu$M and the antagonist, CPPG (Tocris) with a concentration of 300 $\mu$M was used. To block synaptic transmission cadmium chloride (100 $\mu$M) was used.

Single cell microstimulation were performed in juvenile brain explains (see "Explant Preparation"). With a borosilicate glass capillaries neurons were patched (9–15 MOhms) and then stimulated using 0.04–0.5 mA current injections for 500 ms. We choose our microstimulation intensity to match sensory evoked calcium signals in Hb (Fig. 1). A cell was stimulated with up to 60 pulses in total, stimulating 6 times per sweep with 10 sweeps except for one cell that has 7 sweeps (42 pulses in total). To block group III mGluRs, CPPG with a concentration of 300 $\mu$M was used. The calcium activity of Hb was recorded using the 10x objective in the green channel at a frame rate of 10 frames per second.

## Confocal and anatomical imaging

For confocal imaging, *Tg(narp:GAL4VP16;UAS:Syp−GFP-T2A-tdTomato-CAAX)* 14-21 dpf juvenile zebrafish were used. Brains were dissected similar to the explant preparation and then fixed and cleared. For Supplementary Fig. 7, the samples were also stained with DAPI (4′,6-diamidino-2-phenylindole, 1:1000 for 2 h). Anatomical Z-scans were acquired using a Zeiss Examiner Z1 confocal microscope with a ×20 plan NA 0.8 objective or x40 plan NA 1.4 oil objective.

For the example confocal image of the *Tg(narp:GAL4VP16; UAS:Syp−GFP-T2A-tdTomato-CAAX)* brains were cleared using a clearing kit (Binaree) and stained with 4′,6-diamidino-2-phenylindole (DAPI). After fixation, the zebrafish brains were cleared using the Binaree Tissue ClearingTM Kit (#HRTC-012, Binaree, Republic of Korea) using the recommended protocol. On the 3rd day of the clearing process, the zebrafish brains were stained with 1:3000 dilution DAPI for 12 h at 4 °C before being washed with distilled water and then transferred to the mounting solution provided by the kit (#HRMO-006, Binaree, Republic of Korea). The brains were then kept at room temperature (RT) until imaging.

For the RNA detection of mGluR6a we used hybridization chain reaction (HCR) on juvenile zebrafish brains previously described[197]. HCR is combined with a shorten version of the Binaree's clearing protocol. Juvenile zebrafish were euthanized in cold AFW. After euthanasia, juvenile zebrafish were transferred into a 1.5 mL tube containing 200 μL of 4% PFA in dPBS and incubated at 4 °C overnight. Brains were dissected out after washing the samples with 1X dPBS three times at RT for 5 min. Then, the samples were dehydrated and permeabilized with 100% methanol (MeOH) washes for 10 min 4 times

and 50 min 1 time before getting stored at -20 °C overnight. The next day, the samples were rehydrated with a series of graded 1 mL MeOH/dPBS washes for 5 min each at RT: (a) 75% MeOH in dPBS, (b) 50% MeOH in dPBS, (c) 25% MeOH in dPBS, and (d) dPBS. For the clearing stage, the Starting solution from Binaree is used to incubate the samples at 4 °C overnight. Next day, 500 μL of Tissue Clearing Solution B is added and incubated in a water bath at 37 °C for 24 h. Then, the samples were washed with reverse osmosis water while shaking at 30 rpm at 4 °C for 30 min, repeated four times. 500 μL probe hybridization buffer is added, and samples are incubated for 30 min at 37 °C. After removing the probe hybridization buffer, a probe solution (2 pmol of each probe set, prepared by mixing 2 μL of 1 μM stock in 500 μL of probe hybridization buffer at 37 °C) is added. The samples were incubated overnight at 37 °C and washed four times for 15 min each with 500 μL of probe wash buffer at 37 °C. Lastly, the samples are washed twice for 5 min each with 500 μL of 5x SSCT at RT. 500 μL of amplification buffer is added and incubated for 30 min at RT for the amplification stage. Meanwhile, 30 pmol of hairpin h1 and 30 pmol of hairpin h2 are separately prepared by snap cooling (The tubes are heated to 95 °C for 90 s and then cooled to RT in a dark drawer for 30 min) 10 μL of 3 μM stock. The snap-cooled h1 and h2 hairpins are added to 500 μL of amplification buffer at RT, and the samples are transferred into this mixture and incubated overnight in the dark at RT. Samples are washed with 500 μL of 5x SSCT at RT as follows: (a) 2 × 5 min, (b) 2 × 30 min, (c) 1 × 5 min, and with dPBS for 3 ×5 min. The samples are then transferred to 500 μL of the Mounting Solution from Binaree and incubated overnight (12–16 h) in the dark at RT. LSM880 upright Zeiss confocal microscope with 20X (Numerical Aperture: 0.8, Working Distance: 0.55 mm) objectives is used for imaging the samples.

## Generation of Tg(5xUAS:Syp-GFP-T2A-TdTomato-CAAX)[nw19Tg] line
To generate the transgenic line, the *UAS:Syp–GFP-T2A-TdTomato-CAAX* plasmid DNA was used[122]. 2 nL of a mixture of the plasmid DNA (60 pg) and tol2 mRNA (10 pg) was microinjected into one-cell stage embryos, as previously described in Jeong et al.[134]. The injected embryos were raised to adulthood (F0). Germline-transmitted founder zebrafish were identified by breeding with several Gal4 transgenic lines. Stable F1 larvae showing the Gal4 driven GFP and membrane-tethered TdTomato signals were raised to adult zebrafish.

## Ventricular Injections
Larval zebrafish (5 dpf) were mounted in 2% LMP agarose. After solidification of the agarose, the fish were injected with either 1 nL CPPG (5 mM) or vehicle (NaOH, 5 mM) using a glass pipette into the forebrain ventricle. Ventricular injections have been previously described[109,133,134]. After a waiting period of 10 min, the two photon imaging experiment was conducted as explained in section "Two photon calcium imaging".

For the juvenile zebrafish (21 dpf) that were used for behavioral testing the protocol is as follows: Juvenile zebrafish were anesthetized upon transfer to a bath containing 0.01% buffered MS-222 in artificial fish water (AFW). After 10–15 min, upon loss of balance and absence of response to touch stimuli, which was assessed by touching the animal with a soft plastic tip, animals were transferred to a small dish and embedded in 2.5% low gelling point agarose in AFW. When the agarose surrounding the animal solidifies, the immobilized animal was submerged in 0.01% MS-222 in AFW. Next, using a glass micro-capillary CPPG or control (injection volume 1 nL) was injected into the brain ventricle. Then, the fish were freed from the agarose and placed into a dish with fresh AFW to recover. After a waiting time of 10 min, the fish was placed into the behavioral setup for freely swimming behavioral experiments.

## Generation of the mGluR6a mutant
The details for generation and histological characterization of mGluR6a mutants are described in a recent paper where this mutants were first introduced[144]. In brief, Crispr/Cas9 mutagenesis was performed as previously described in Schlegel et al.[198]. CRISPR target sites of the mglur6a gene selected using CHOPCHOP (chopchop.rc.fas.harvard.edu). Targets sites were: CGAGGAGGTCCAATCTAACC (T7) and GACCAGGAGGACGTGGCTGA (Sp6). Forward and reversed oligonucleotides (pT7-gRNA_fwd: CAGCTATGACCATGATTACG; pT7-gRNA_rev: AAAAGCACCGACTCGGTG; pSp6-gRNA_fwd: ATTTAGGT-GACACTATA; pSp6-gRNA_rev: ATTTAGGTGACACTATA) were cloned into pT7- or pSP6-sgRNA as described in Jao et al.[199]. sgRNAs were transcribed using MEGAscript Sp6 kit (Ambion, Austin, TX, USA) and MEGAshortscript T7 kit and purified using the MEGAclear kit (Ambion, Austin, TX, USA). The resulting mutant has a 103 bp deletion with no detectable mGluR6a protein labeling[144].

## Behavioral assays
The novel tank test was conducted in a custom behavioral set up that allowed to simultaneously record 6 fish in 6 separate arenas (11.5 cm × 11.5 cm × 1.5 cm) from the side. In short, fish were introduced to the arena in the first 5 seconds of the experiment where the bottom row of fish was introduced first, following the top three fish. The fish were then tracked for their x and y position and the data later analyzed with custom MATLAB scripts. The total length of the experiment was 30 min. The time point used for the analysis was the first two minutes. The mechanical vibration assays were conducted in an Zantiks LT set up with 6 arenas (10 cm × 11.5 cm × 3 cm) allowing to perform the experiment with six fish simultaneously while recording their position from the side. The fish were individually tracked (x and y position) and the data later analyzed in custom MATLAB scripts. For the mechanical vibration experiment, the fish were habituated to the arena for 20 min. Then after a 4 min baseline, 15 mechanical vibrations were delivered to the 6 arenas simultaneously with an interstimulus interval of 1 min, followed but a 4 min baseline period. The Light/Dark Transition experiment was conducted in a Zantiks MWP set up using 6-well plates (VWR) that allow tracking of the animal movement from above (Fig. 7E–H). In short, fish were introduced to the well plates and placed in the set up. After a habituation period of 15 min in light, the behavioral protocol started with the dark condition (5 min) and then back to the light condition (5 min). This was repeated 10 times. Moreover, we investigated novel-tank like defensive behaviors (Supplementary Fig. 10A, B) in controls versus CPPG-injected animals using the Zantiks MWP setup viewing animal behavior form above. To do this, we compared animals swimming speed during first 3 min of introduction to tank versus after a period of adaptation minutes between 7–10. This comparison revealed a significantly larger speed recovery in control animals, when compared to CPPG injected animals keeping their swimming at low speeds and do not recover. Finally, we investigated how control versus CPPG-injected zebrafish alter their swim speeds up on mechanical vibrations (Fig. Supplementary 10C, D) using the Zantiks MWP setup viewing animal behavior form above. To do this, 10 mechanical vibrations were delivered to freely swimming control and CPPG-injected zebrafish, after the initial and transient increase of swim speed in both groups, CPPG-injected group showed a significantly larger reduction of speed. The fish were individually tracked (x and y position) and the data later analyzed in custom MATLAB scripts.

Experiments conducted with juvenile mGluR6a mutants were done blindly by using a heterozygous incross. After the experiment, the fish were genotyped using real-time quantitative polymerase chain reaction melt-curve analysis[200]. Using the forward primer GCTTGAGCATAAAACTCTAATTC and the reverse primer CAGAGGATGCACATTATATTTC.

## Transcriptomics analysis
The scRNAseq datasets from zebrafish larvae[113] and mouse[117] were analyzed using R and the Seurat R Package v3[201]. Non-neuronal cells in

larva were removed based on the expression of *snap25a*, for mouse they were removed based on the expression of *stmn2, thy1* and *snap25*. To identify dorsal and ventral Hb in the zebrafish, *kiss1* and *aoc1* were used, respectively. For the mouse MHb was identified with *tac2* and LHb was identified with *pcdh10* marker genes. For zebrafish spatial transcriptome sections, six-month old zebrafish ($n = 2$) were euthanized in cold water, and their brains were dissected. Brains were fixed in PAXgene Tissue FIX, stabilized in PAXgene Tissue STABILIZER, and cryo-sections were collected on precooled coverslips. Detailed protocol is described in D'Gama et al.[126].

Mouse habenula single cell and spatial transcriptome data is obtained from Yao et al.[54] and spatial expression and gene imputation images for group III mGluR expression in Hb are generated using the Allen Institute for Brain Science – Brain Knowledge Platform(https://knowledge.brain-map.org/abcatlas?defaultProjectId=LVDBJAW8BI5YSS1QUBG).

### Ages of fish
Juvenile zebrafish have been shown to perform cognitively demanding behaviors[62,67,146–150] which is why we used this age group for most of our experiments. 3-4 weeks old juvenile zebrafish were used for preparing the whole-brain explant in electrophysiology and pharmacology experiments, as well as in vivo calcium imaging of mGluR6a mutants and all behavioral experiments. For the in vivo ventricular injections, larval zebrafish (5 dpf) were used due to national ethical limitations in performing such in vivo ventricular injections in juvenile zebrafish.

### Data analysis
Whole brain explant two-photon microscopy images were aligned using methods described in previous work[109,110,118,119]. The in vivo recordings were aligned using suite2p[202]. After alignment all the recording were treated the same. The recordings were visually inspected for any remaining motion z-directional drift and discarded if any motion artifacts remained. Using a template matching algorithm[107,110,119], regions of interests (ROIs) corresponding to neurons were detected and visually confirmed. Pixel belonging to each ROI were averaged over time to calculate the time course of each neuron. Then the fractional change in fluorescence relative to the baseline ($\Delta F/F$) was calculated. For sensory stimulation trials, the signal was normalized to a 5 s pre-stimulation period. For the spontaneous period in vivo and the whole brain explant experiments an average over moving window of 6 min was used. For the in vivo recordings the calcium signals were smoothed using filtering previously described[128].

For the calcium imaging while stimulating a single cell, the 10 (or 7) sweeps were combined into a continuous recording and then motion corrected using NoRMCorre[203]. Recordings with significant motion artifacts observed after manual inspection were discarded. In the remaining recordings, a ROI was drawn around the Hb neuron that was patched. The rest of the cell detection as well as the signal extracted using CNMF-E[204]. In addition, the cell detection accuracy was manually inspected and detected ROIS that did not relate to an actual cell were rejected. The patched neuron was manually isolated from the rest of the neurons. The extracted calcium traces were smoothed using a gaussian convolution and normalized using the mean and baseline from the pre-stimulation period.

All the following analysis was performed in MATLAB.

Responding cells (light or mechanical vibrations) were calculated by comparing the response average over a 10s-time window after stimulus onset to the baseline activity. Excitatory responses exceed two times the standard deviation (STD) plus the mean of the baseline duration. In addition, the same analysis was performed for the CPPG-injection experiment using 1, 3 and 4 STD plus the mean see Supplementary Fig. 4, which reveals that the effect is not based on a specific threshold. Inhibitory responses are below the STD minus the mean of the baseline period. For the single cell stimulation experiments, a not

tailed Wilcoxon signed-rank test was performed using median values from the pre-stimulation period and the first two seconds post-stimulation. If a neuron was responsive ($p < 0.05$), it was considered inhibited if the median of the post-stimulation period was below the median of the pre-stimulation period and excited if the post-stimulation median was above the pre-stimulation median. Recordings in which the stimulated cell did not show statistically significance (excited) were rejected.

Affected cells to the antagonist (CPPG) and agonist (L-AP4) were calculated in a similar way, with CPPG-affected neurons are exceeding two times the STD plus the mean of the baseline period (5 min before wash in) during the drug period (minute 13 to 18 of the experiment) and L-AP4-affected neurons responses are smaller than the mean minus the STD of the baseline period (2 min before wash in) during the drug period (minute 26 to 28 of the experiment).

Delineation of the forebrain regions was done manually according to anatomical landmarks as described previously[110,131,132].

The response vectors to light or mechanical vibration were vectors of the average response of each neuron during the response period. To find their similarities the two vectors were correlated with each other using Pearson's correlations. As a complementary measure of similarity, we also used cosine similarity (in Supplementary Fig. 4J), which reports the angle between multi-neuronal vectors representing two different sensory stimuli and hence do not rely on response amplitudes.

To understand the response observed in Hb neurons to the combined presentation of mechanical vibration and light, we calculated the interaction index which is based on previous work[142,143] and also knowns as the "interactive index" in the original publication. It is calculated as followed:

$$interaction\ index = \frac{R_{Light\&Vib} - max(R_{Light}, R_{Vib})}{max(R_{Light}, R_{Vib})} * 100$$

The responses (R) to light, mechanical vibration or the co-stimulation (light and vibration) are average over the response period (10 s after stimulus onset). An interaction index of above 100 represents an "super-additive" response where the response to the co-stimulation is at least twice as big as the maximal response to the individual stimuli. On the other hand, a interaction index of below zero represents a response to the co-stimulation that is smaller than the max response of the individual stimulation called "response depression". An interaction index between 0 and 100 is named "sub-additivity".

To find the most dorsomedial Hb neurons for mGluR6a mutant imaging experiments, the y-position of the neurons was normalized to the middle point (mean). The z-position was normalized to the highest point. Neurons were then ranked according to the absolute sum of the normalized y and z position. Small sums represent neurons that are dorsal and close to the midline. The 40% percent neurons with the smallest sum were classified as the most dorsomedial neurons.

To split the larval Hb from the injection experiments into dorsal and ventral Hb, the same normalization of the z-position was used. We then used a threshold of 40 µm to determine which neurons are dorsal (below 40 µm distance from the top) and ventral (equal to or above 40 µm distance from the top).

Midbrain neurons posterior to the habenula are identified based on anatomical landmarks in larval zebrafish.

For the behavioral experiments the x and y positions of the animals were tracked. The heatmaps represent the average density of zebrafish positions. To calculate the distance from the bottom, the y-position of the animals were normalized to the dimensions of the tank. The change in depth was calculated by fraction of the y-position two seconds before the mechanical stimulation and the y-position after the stimulation. The distance change is the fraction of the swimming distance of the fish in x-y direction in 1 s bins five seconds before the transition and the rest of the duration.

## Quantification and statistical analysis

MATLAB was used for the statistical analysis. *P*-values are represented in the figure legends as *$p < 0.05$, **$p < 0.01$, ***$p < 0.001$. All analysis was performed in MATLAB, R and Fiji.

## Reporting summary

Further information on research design is available in the Nature Portfolio Reporting Summary linked to this article.

## Data availability

All raw data generated in this study have been deposited in the sigma2 data base under the accession code https://doi.org/10.11582/2025.tdq8m63j. Source data are provided as a Source Data file. Source data are provided with this paper.

## Code availability

All the codes used in this study are available at Zenodo with the following https://doi.org/10.5281/zenodo.15752614.

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

## Acknowledgements

We thank M. Ahrens (HHMI, Janelia Farm, USA), K. Kawakami (NIG, Japan), and H. Okamoto (RIKEN-CBS, Japan) for transgenic lines; Y. Yoshihara RIKEN-CBS, Japan for sharing the *UAS:Syp–GFP-T2A-tdTomato-CAAX* plasmid, S. Eggen, O. Mc Innes, V. Nguyen, and our fish facility support team for technical assistance; the Yaksi lab members for stimulating discussions. This work was funded by an RSO grant from Norwegian University of Science and Technology (E.Y., A.M.O.), Boehringer Ingelheim Fonds (S.K.J.), Koc University Neuroscience Master's Program fellowship (YIC), ERC starting grant 335561 (E.Y.), NFR FRIPRO research grants 314189 (N.J-Y), 239973 (E.Y.), 314212 (E.Y.) and RCN Centres of Excellence scheme, project number 332640. Work in the E.Y. laboratory is funded by the Kavli Institute for Systems Neuroscience at Norwegian University of Science and Technology.

## Author contributions

Conceptualization, A.M.O. and E.Y.; Methodology: A.M.O., N.F., B.S., I.J., E.D.D., A.E., S.C.F.N., F.H., R.B., S.K.J., N.J-Y. Analysis A.M.O., N.F., Y.I.C.C., A.K.M. and E.Y.; Investigation, all authors; Writing A.M.O. and E.Y.; Review & Editing, all authors; Funding Acquisition and Supervision, E.Y.

## Funding

 Olavs Hospital - Trondheim University Hospital).

## Competing interests

The authors declare no competing interests.
