## [Transparent Peer Review file · Nature Communications]

Inhibition mediated by group III mGluRs regulates habenula activity and defensive behaviors

Corresponding Author: Dr Emre Yaksi

Version 0:

Reviewer comments:

Reviewer #1

(Remarks to the Author)

This study investigated the role of inhibitory glutamate receptor signaling in the habenula using larval and juvenile zebrafish. Inhibitory signaling from glutamate receptors is a known phenomenon, yet poorly understood especially in the context of circuit function and animal behavior. Moreover, the habenula is a highly conserved midbrain structure with crucial roles in limbic regulation and multi-sensory integration. This study also used a variety of powerful techniques including spatial transcriptomics, calcium imaging, electrophysiology, pharmacology, and genetic perturbation to address targeted questions regarding the role of inhibitory mGluR III signaling in habenula function. Therefore, the study approaches important questions and addresses gaps in knowledge that will likely be of broad interest across scientific fields. The combined array of utilized techniques, especially in older juvenile larval, adds technical innovation to the study as well. Overall, I think this is a highly appropriate study to be published in Nature Communications. OF particular interest is finding that glutamate receptor inhibitory signaling is important for intra-habenula selectivity and multi-modal sensory processing – experiments that were executed very nicely. That being said, I do think there are several key points that should be addressed to strengthen the conclusions and some minor points to improve clarity.

Major:

1. Implementing the mGluR6a CRISPR KO is great yet having this mutant begs for some functional analysis. Performing at least a single calcium or electrophysiology experiment, similar to prior experiments in the manuscript, would ideally recapitulate pharmacologic antagonist results. The fact that behavior is modified suggests that habenula function should be perturbed, yet experimentally validating this would be highly rigorous and powerful.
2. The others state the scRNAseq may not have the resolution to identify GPCRs, yet stay with the same scRNAseq targets for spatial transcriptomic analysis. It would be worth looking at habenular expression of a few other mGluRs, at least the a/b versions of the current receptors to verify specificity. This may also be worth looking at in mGluR6a mutants to provide some evidence of limited or no compensation of closely related receptors or known habenula expressed receptors.
3. For the CRISPR mutant it's stated that there is a lack of protein expression with a 'data not shown'. Since this data is available it would be worth adding as a supplemental.
4. The resolution of the habenula synaptic connectivity does not allow for a rigorous or convincing conclusion and better images or membrane labeling would help solidify this important result. Notably, this is challenging with the utilized tools as so many versions of red fluorescence fluorophores are notorious for aggregation.

Minor:

1. Can latency of excitatory and inhibitory responses be quantified from any of the current data, such as in Figure 1 or 5. This analysis could provide support from within this current study for points made in the discussion regarding the rates of excitatory and inhibitory responses.
2. Figure 1A the z-axis is hard to appreciate. Providing two orthogonal views may provide more context
3. Whenever average calcium traces are shown it would be helpful to show SEM for a reader to better appreciate the range of observed responses.
4. 2nd sentence in abstract is confusing
5. Spatial transcriptomics (is this the molecular cartography?) could be better described in the methods and include the N's

performed

6. Providing some labeling of positions/regions in the spatial transcriptomics image would be helpful

7. In Figure 2E middle panel are the different treatment conditions performed sequentially on the same individual. The line graph format gives this impression and should be clarified.

8. Figure 3B what is the grey area on the graph?

9. In Figure 5D how many different habenula neurons were stimulated. Do you think number of responsive neurons would change with stronger stimulation of the target neuron?

10. It could be interesting to see if pharmacologic manipulation of G-protein α function would recapitulate direct receptor manipulation. I don't think this experiment needs to be done yet may be worth considering/discussing.

(Remarks on code availability)

Reviewer #2

(Remarks to the Author)

The study by Ostenrath et al. investigates the inhibitory action of Glutamate receptors, specifically the group III mGluRs, in the physiology of the zebrafish habenula and behavior. The authors provides important insights into the so-far understudied function of group III mGluRs and show their importance not only in the inhibitory neuronal responses to sensory stimuli, but also in animal behavior. It underscores the diverse action of Glutamatergic transmission through this group of mGluRs. The experiments are rigorously performed and the results are presented in a well-organized manner. On the other hand, my major concern is about an uncertainty about whether the group III mGluRs-dependent inhibition is directly mediated within the habenula (Hb) circuit or indirectly mediated by non-Hb neurons (as detailed below). This issue is mainly caused by the lack of specificity of pharmacological and genetic manipulation of the mGluRs. Anatomical analyses suggest some potential neural circuit inside the Hb, however, this anatomical finding is not integrated with the functional observations. Overall, the authors' interpretation that this inhibition is mediated through circuits within the Hb is not sufficiently supported by data in this manuscript. Another main concern is that the behavior tests that they employed to show defects in defensive behavior in mGluR6a have not been shown to require Hb activity, so the link between the mGluR6a, Hb and the behavioral effect is rather weak.

Major points:

1) Pharmacological manipulation of the group III mGluRs are performed by either bath application (juvenile) or ventricular injection (larvae) of antagonist (CPPG) and agonist (L-AP4). Both approaches act not only on the group III mGluRs expressed in the Hb, but also on those expressed anywhere in the brain. The authors show that the group III mGluRs are strongly expressed in the juvenile dorsal Hb compared to the forebrain (Fig. 2C,D), but no evidence is provided for whether they are expressed in different parts of the brain other than the Hb and forebrain. Previous paper (Huang et al., PLOS One, 2012) showed that mGluR6a is strongly expressed in the Hb, as well as other brain areas (such as optic tectum and midbrain-hindbrain-boundary) at larval stages, suggesting that CPPG can act on neurons outside the Hb. In line with this, CPPG application increases activity of Hb and olfactory bulb, as the authors describe, but an area posterior to the Hb seem to show some increased activity upon CPPG administration, suggesting that neurons located outside of the Hb can be manipulated by CPPG (Fig. 3A). Therefore, one cannot exclude the possibility that the pharmacological effect is primarily mediated by non-Hb neurons, which in turn affect Hb neurons. Such indirect effects should be carefully discussed. Although the authors do acknowledge this by saying that "This cross-modal inhibition in Hb is by multi-synaptic inhibition through Hb-interpeduncular nucleus (IPN) connections onto presynaptic terminals of Hb neurons (115)" as a possibility, the overall tone of the manuscript is that the site of group III mGluRs' action is on the Hb. For instance, "Our results revealed prominent inhibitory connections across Hb neurons, which are largely reduced by blocking group III mGluRs" (Introduction) and "We revealed that glutamatergic Hb neurons can directly communicate with each other" (Discussion).

2) In the example shown in Figure 5D, the electrically stimulated cells show extremely strong activity of DF/F more than 60. This does not seem to be physiological by comparing the Hb responses in Fig. 1A. Despite this strong activity in the simulated cell, the evoked activity in the Hb are rather small (around 2 DF/F). This could suggest that the connection between the simulated cells and the surrounding cells are rather weak or indirect. The authors should perform this experiment in more physiological conditions.

3) The microstimulation experiment in Fig. 5D uses the whole brain explant. Related to my point 1), the evoked activity in the Hb could be indirectly generated by the Hb-IPN pathway. It would be interesting if the authors perform this experiment using the explants only containing the Hb, although this might be technically difficult.

4) Interestingly, the authors describe that dHb neurons have presynaptic terminals in the Hb neuropil region (Fig. 5G, H). This is quite new and exciting, however I cannot see a direct innervation from the soma to this neuropil region from the image provided. Can they show a neurite that originate from cell bodies of dHb neurons that terminate in the Hb by looking at the tdTomato signal? This could be done by 3D projection or tracing of the neurites. In addition, can they show that the group III mGluRs are expressed near this terminal region? This would be a prerequisite for the glutamate released by dHb neurons to induce inhibition in the postsynaptic Hb neurons.

5) It would be nice if they can show that the group III mGluRs-dependent inhibition in the Hb with the knowledge of Hb

anatomical innervation pattern. For example, if they microstimulate one of the dHb neurons labeled in Tg(narp:GAL4VP16), do they observe the inhibition in putative postsynaptic cells located close to the Tg(narp:GAL4VP16)-positive neurites? This way they can correlate the spatial specificity of the intra-Hb projection and the evoked inhibition.

6) Fig. 2 shows that the group III mGluRs are more strongly expressed in dHb than vHb at larval stage. In the CPPG experiments presented in Fig. 4 and 6, it is expected that the effect of CPPG is stronger for the dHb compared to the vHb. This can be tested by genetically labeling dHb or vHb neurons and repeating the same experiments.

7) The mutant model of mGluR6a is lacking mGluR6a protein in any parts of the brain. The authors cannot conclude that the observed behavioral phenotypes are caused by the Hb, unless they present evidence that the mGluR6a is not expressed anywhere in the body. As I mentioned in point 1) above, the mGluR6a is expressed not only in the Hb, but also in the optic tectum and midbrain-hindbrain boundary, suggesting that the Hb-independent effect cannot be excluded. The authors may be aware of this issue, but please explain them more explicitly.

8) The authors state that "Previous research has shown that Hb is involved in various defensive behaviors (57-64, 66, 92, 101-103)". This set of references includes studies from both mouse and zebrafish. It is unclear why they chose to test behavior that are different from those that have been already shown to involve the Hb. The behavior tests that they examined in this study generally address fish response to aversive stimuli, but none of them have been shown to require Hb activity. From the data, they can only conclude that the mGluR6a gene is involved in particular types of defensive behavior, but it is inconclusive whether or not the effect of mGluR6a is mediated by its expression in the Hb.

Minor points

1) The authors' group previously showed that Hb responds asymmetrically to the light and odor stimuli (e.g. Fore et al., Science Advances, 2020). The Fig. 1A does not show clear asymmetry as previously reported. Why is it the case? Also, in this figure, there are different shades of orange and blue dots, but the explanations are not given. Please clarify.

2) On page 4, "Notably, we identified topographically distinct expression of mGluR4, mGluR6a, mGluR8b in the zebrafish dHb (Figure 3C)": do they mean Fig. 2C, not Fig. 3C?

3) Fig. 3D, G: the y-axis label should be "affected" but not "effected".

4) In Supplementary Fig. 4, they checked whether the mGlu6a mutants recapitulate the pharmacological alterations. Although they mention that "These results are very similar to the changes in sensory representations we observed upon blocking group III mGluRs via ventricular CPPG injections (Figure 4)", but this statement is not correct because some phenotypes are not actually recapitulated and also they compare juvenile vs larvae. The authors should describe this more carefully and precisely.

5) Fig. 7 legend: the authors might want to use "light and dark transition", but not "dark and light transition".

6) Fig. 7E: There are two scale bars for time. Which one is correct?

7) In the Discussion, they discuss conservation between zebrafish and mouse. Since very little conservation was found in terms of the expression patterns in this study (Fig. 2), this discussion seems not relevant in this case.

8) The discussion about the time scale of mGluR6a seems unrelated to the data presented in this manuscript.

(Remarks on code availability)

Reviewer #3

(Remarks to the Author)

(Remarks on code availability)

Reviewer #4

(Remarks to the Author)

Overview:

This manuscript addresses how glutamatergic signaling, mediated by metabotropic glutamate receptors (mGluRs), can affect the neural physiology, circuit function, and the behavior. Specifically, the authors use the habenula of larval and juvenile zebrafish to explore the roles of group III metabotropic glutamate receptors.

They first show that diverse sensory inputs can drive both excitation and inhibition in habenula neurons. There are more excited neurons than inhibited neurons for vibration and light stimuli, while there are more inhibited neurons in response to odors. After finding that group III mGluRs are expressed in the habenula, they show that an agonist hyperpolarizes Hb neurons in general, and that an antagonist depolarizes them. This supports the notion that glutamate, acting through group III mGluRs, has a generally inhibitory effect in the habenula. Calcium imaging in explants suggests that this hyper- and depolarization has the effect in increasing and decreasing habenular activity, respectively. Moving back in vivo, they show that Hb neurons have greater excitation and less inhibition with to visual and vibration stimuli when mGluRs are blocked, with the knock-on effect that more neurons show responses (above 2SD) to both types of stimuli. This is followed by an analysis of correlations between pairs of Hb neurons at a range of distances, with and without antagonist, from which the authors draw the conclusion that mGluRs are responsible for inhibition that is manifested as negative correlations at longer distances. Such inhibition could support competition between sensory stimuli, so the authors then look at the (mostly suppressing) interactions between responses to combined visual and vibration stimuli. Antagonist shifted these effects subtly away from suppression between stimuli in the responses of Hb neurons. Finally, they showed that mutants for one of the group III mGluRs have similar activity patterns to those in fish treated with antagonist, and that these animals have enhanced defensive behavior.

Overall, I find the work interesting. It provides a satisfying example glutamatergic inhibition, different elements of this signaling through electrophysiology, calcium imaging, and behavior. I do not view the outcomes as groundbreaking, but think that they will be of interest to a wide audience of neuroscientists, principally those interested in glutamate and those interested in the role of the habenula. I have presented several major comments below. The most important of these is captured in comments 3 and 4. I am concerned that the generally increased excitability of Hb neurons treated with the group III mGluR antagonist explains most or all of the results, and that inhibition is not convincingly implicated by these analyses as currently presented.

Major comments:

#1

I find the evidence for mGluR expression in the mouse habenula to be weak. It is based on the analysis of a published dataset, and identifies robust transcript for only two out of four receptors that were studied. The abundance of these transcripts in the Hb versus other brain regions is not described, and the relevance of the expression level values is not put in context. Overall, these data are much weaker than the equivalent zebrafish data. I assume that these results are included to generalize the conclusions drawn from the zebrafish work, but I believe that they lower the standard of the paper's data, and I would suggest removing them.

#2

The nature of the sensory stimuli applied in Figure 1 is important. Odor, in particular (although vision also), can be either attractive or threatening, and without testing a range of stimuli, it is difficult to gauge whether all odors lead principally to inhibition or whether different odors lead to different balances of excitation and inhibition. This becomes an important consideration when behavioral responses enter the story.

In general, insufficient technical detail is provided about the sensory stimuli. What is the lux change in the imaging chamber when the red LED is illuminated? Where is the solenoid located and what is the volume of the vibration stimulus within the chamber? Did you try a range of stimuli before deciding that these were the appropriate ones for testing individually and a multisensory combinations? Since rather careful analyses of responses to both visual and vibrational stimuli exist in the literature, this information is important for putting the current results in context.

#3

The claims made around the data in Figure 4 need more consideration. It is convincing that there is more activation by stimuli in Hb neurons when the antagonist is present. This effect spans visual and vibration stimuli. Therefore, it is unsurprising (maybe inevitable) that more neurons will exceed 2SD with each stimulus. This is presented as evidence that Hb neurons become less selective to stimuli, when all that has been shown is that they are more excitable. The claim regarding selectivity depends on there being something sacred about 2SD, which there is not.

Perhaps a claim of less selectivity could be made if the ratio of visual to vibrational stimuli changed (to become more similar to one another with antagonist), but this would have the caveat that the calcium signal is not a linear representation of activity. All things considered, I do not buy the interpretation that they are less selective; I am only convinced that they are more excitable.

#4

As with the argument about selectivity in Figure 4, the claims about inhibition and distance presented in Figure 5 may be overlooking a simpler explanation. If Hb neurons are simply more responsive with CPPG, then positive correlations may increase since synchronized ensembles might be larger or stronger. I do not like how the data in 5E and F are presented given this possibility. Given that the blue data in E and the grey data in F are the same datapoints, I would prefer to see this treatment presented only once. The current display and description give the false impression that they are data from different experiments. Then, I would like to see both excitation and inhibition for both aCSF and CPPG (four bars). This will clarify for the reader that a single experiment has been done, and will distinguish between the specific loss of inhibition in CPPG versus a generally more excitable (and correlated) network. The excitation data from the CPPG treatment are needed to make this judgement.

Minor comments:

#5

In Figure 2 E and F, is the experimental n given the number of neurons or the number of explants? If neurons, what was the number of explants?

#6

I'd like to see a panel like 3B for L-AP4 so that the timecourse and magnitude of the agonist can also be appreciated.

#7

In Figure 3, was co-labeling with DAO really necessary to identify the vHb? I had thought that there was going to be an interpretation of what role DAO+ neurons might specifically be doing, but if they're just to show which neurons are ventral, the 3D image should have been sufficient to show this. I feel like the DAO angle is a little distracting.

(Remarks on code availability)

Version 1:

Reviewer comments:

Reviewer #1

(Remarks to the Author)

The authors have sufficiently addressed my concerns and comments. This is a compelling story supported by some very nice experiments.

(Remarks on code availability)

Reviewer #2

(Remarks to the Author)

The authors have adequately addressed most of my questions, by adding new data and revising the text. However, I would like to highlight three unresolved points that should be addressed before publication.

-Re. Response 18: Thank you for addressing this point. The new images provided in Fig. 5 now better illustrate the presynaptic marker in the neuropil of the Hb. In the authors' response, they discuss the possibility that inhibition may be mediated between Hb dendrites, possibly by the dendro-dendritic synapses. Although this is speculative at this stage, it would be valuable to explicitly mention this interesting possibility in the manuscript, and I encourage the authors to do so. I feel that the images in Fig. 5G deserve further explanation.

-Re. Response 21: I agree that the behavior assays that they performed are well-established for testing defensive and adaptive behaviors. However, to my knowledge, the behavior assays that the authors used here have not been shown to be mediated by the Hb (Bühler and Carl, 2021, <https://doi.org/10.3390/biom11020324>). For example, silencing dHb in zebrafish has been shown not to alter performance in the novel tank diving test (Chou et al., 2016, DOI: 10.1126/science.aac9508). Therefore, I still think that it is unclear whether the behavioral defects in the mGluR6a mutants are due to alterations in Hb function. The new data in Fig 7G nicely link the site of action of the group III mGluRs. Can they test whether the CPPG injection also phenocopies the other behavioral tests (novel tank assay and mechanical vibration)? These data would be valuable, as it could further support the connection between the group III mGluRs, Hb and the behavior.

-Re. Response 28: The new data on the spatial transcriptomics of mouse mGluR8 (presented in Supplementary Figure 2D, E) is somewhat confusing. The data here shows that mouse mGluR8 is expressed more strongly in the medial Hb (MHb), while the data in Figure 2B indicates that the expression of mouse mGluR8 is stronger in the lateral Hb (LHb). How can this discrepancy be reconciled?

(Remarks on code availability)

Reviewer #3

(Remarks to the Author)

(Remarks on code availability)

Reviewer #4

(Remarks to the Author)

There are improvements in this manuscript, but still some important concerns that have not been addressed. These are detailed below, according to the numbers of the original comments.

Major comment 1:

I think that the addition of new data makes the mouse study less cursory, but I still believe that its inclusion detracts from the paper's focus and does not support an argument for a key role for the habenula (given the broad expression for the three mGluR genes studied). While I continue to disagree about the value of these data, I don't see them as an obstacle to publication.

Major comment 2:

I see the new supplementary figure 1 as a valuable addition. It can be mentioned quickly in the text, so as not to interfere with the manuscript's focus, but still addresses the concern about the valence of olfactory cues. I would suggest leaving it in the manuscript's supplementary section.

My concern about stimulus parameters has not been addressed.

I still believe that some actual measurements of the stimuli are in order. Saying the wattage or voltage across the LEDs or solenoid does not tell the reader what the animal is experiencing. Citing past work does not solve this problem unless measurements are reported in those papers.

Major comment 3:

This comment has not been meaningfully addressed.

Given that 2SD is arbitrary (if frequently used), it would be good to show results from a few different thresholds to show that the results being reported are general, rather than being dependent on the use of a specific threshold.

The threshold used is less important than the interpretation, though. A general increase in the population will both have the effect of producing more multimodality (at a given threshold) and also the effect of increasing the correlations across treatments. If more neurons across the Hb are active with antagonist or in the mutant, and if that activity is tightly correlated to stimulus presentations, it will result in higher correlation values, even if there is no biologically meaningful lack of selectivity taking place. All of these effects could be the results of a simple increase in responsiveness.

In Figure 4H and Supplementary Figure 8J, it appears that the unimodal and multimodal populations add up to 100%. As such, it is unnecessary (I would say inadvertently misleading) to represent them as separate data, especially with the inclusion of apparently different statistical tests on the same data. I would suggest presenting these as stacked bar graphs to avoid confusion.

Major comment 4:

I am satisfied with these results given the two panels that have been added, and I see the authors' points about the need to show the results separately.

Minor comments 5-7:

These have been adequately addressed.

(Remarks on code availability)

Version 2:

Reviewer comments:

Reviewer #2

(Remarks to the Author)

The authors have sufficiently addressed my concerns by adding new data and revising the text. With all the revisions presented, the manuscript has significantly improved.

(Remarks on code availability)

Reviewer #3

(Remarks to the Author)

(Remarks on code availability)

Dear Editors and Reviewers,

We are delighted to see the reviewers' enthusiasm for our study and greatly appreciate their compliments on the quality of our data and results. We are also grateful for their constructive and thoughtful feedback, which has certainly helped improve the clarity and value of our study.

After carefully considering all the comments, we have prepared a revised manuscript that addresses them to the best of our ability. You will find our detailed responses below, where we explain how we addressed each comment through new experiments, analyses, clarifications, textual adjustments, several new main figure panels and 4 new supplemental figures. To help with the revision, we highlighted the updated parts of our manuscript using yellow highlights. We are confident that our manuscript is now significantly strengthened and more comprehensive. We thank all reviewers for their suggestions.

All reviewers' comments are included below in red, and our responses are provided in black.

We look forward to hearing your feedback on our revised manuscript.

With kind regards,

Emre Yaksi

Anna Maria Ostenrath

Reviewer #1 (Remarks to the Author):

This study investigated the role of inhibitory glutamate receptor signaling in the habenula using larval and juvenile zebrafish. Inhibitory signaling from glutamate receptors is a known phenomenon, yet poorly understood especially in the context of circuit function and animal behavior. Moreover, the habenula is a highly conserved midbrain structure with crucial roles in limbic regulation and multi-sensory integration. This study also used a variety of powerful techniques including spatial transcriptomics, calcium imaging, electrophysiology, pharmacology, and genetic perturbation to address targeted questions regarding the role of inhibitory mGluR III signaling in habenula function. Therefore, the study approaches important questions and addresses gaps in knowledge that will likely be of broad interest across scientific fields. The combined array of utilized techniques, especially in older juvenile larval, adds technical innovation to the study as well. Overall, I think this is a highly appropriate study to be published in Nature Communications. Of particular interest is finding that glutamate receptor inhibitory signaling is important for intra-habenula selectivity and multi-modal sensory processing – experiments that were executed very nicely. That being said, I do think there are

several key points that should be addressed to strengthen the conclusions and some minor points to improve clarity.

Major:

1. Implementing the mGluR6a CRISPR KO is great yet having this mutant begs for some functional analysis. Performing at least a single calcium or electrophysiology experiment, similar to prior experiments in the manuscript, would ideally recapitulate pharmacologic antagonist results. The fact that behavior is modified suggests that habenula function should be perturbed, yet experimentally validating this would be highly rigorous and powerful.

Response 1: We thank the Reviewer 1 for complimenting the quality and significance of our work. To answer this comment above: In fact, we have performed calcium imaging experiments with mGluR6a mutants as Reviewer 1 has asked and observed similar features of altered neural excitability and selectivity of Hb (from here on we will use Hb to abbreviate Habenula) neurons located in areas expressing mGluR6a. However, these experiments were perhaps overlooked as they were placed in Supplementary Figure 8 in our revised manuscript. If the editors and reviewers prefer, we can move these results as main figures. We also tried to better clarify that these functional imaging experiments were performed in mGluR6a mutants. Moreover, during this revision process, to better bridge the impact of group III mGluR pharmacology in Hb function (Figure 4-5) and animal behavior, we performed complementary behavioral (light-dark transition assay) experiments with CPPG-injected zebrafish (Figure 7G-H). These new pharmacological perturbation experiments resulted in behavioral phenotype (Figure 7G-H) very similar to mGluR6a mutants (Figure 7E-F). These new results are presented in our revised manuscript.

2. The authors state the scRNAseq may not have the resolution to identify GPCRs, yet stay with the same scRNAseq targets for spatial transcriptomic analysis. It would be worth looking at habenular expression of a few other mGluRs, at least the a/b versions of the current receptors to verify specificity. This may also be worth looking at in mGluR6a mutants to provide some evidence of limited or no compensation of closely related receptors or known habenula expressed receptors.

Response 2: We did not make any statement about the resolution of scRNAseq in our study. Yet, spatial transcriptomics method is better in identifying rare transcripts such as GPCRs. Nevertheless, both in zebrafish and mouse scRNAseq data we have observed significant presence of group III mGluRs. However, we would like to highlight that we choose our spatial transcriptome targets not based on scRNAseq data from zebrafish, but based on extensive *in situ* stains for all mGluRs in juvenile and larval zebrafish data presented in Haug et al. 2013 (<https://doi.org/10.1002/cne.23240>). Our spatial transcriptome results align very well with those *in situ* stains in this past paper, which also shows the expression of other group III mGluRs as reviewer 1.

Nevertheless, as requested by the reviewer we now generated a new data figure (Supplemental Figure 2B) that shows the expression of both mGluR6a and mGluR6b. These new results revealed the lack of expression of mGluR6b in zebrafish habenula, despite prominent expression of mGluR6b in the olfactory bulb (Supplemental Figure 2A).

A recent paper from Haug et al. 2024 (<https://doi.org/10.1167/iovs.65.12.44>) have confirmed that mGluR6a knock out fish, which we used in this study, lacked the expression of relevant

mGluR6a protein. We agree that potential genetic compensatory mechanisms can be at play in any mutants, across several different genes, in all studies. However, we argue that the similarities of our functional (Figure 4 vs Suppl Figure 8) and newly presented behavioral changes (Figure 7E-F vs Figure 7G-H) in our mGluR6a mutants and CPPG-injected fish, strongly indicate that such acute (pharmacology) and chronic (genetic) perturbations generate overlapping results. These new results suggest minimal genetic compensation at the level of animal behavior. These results are now added in the revised manuscript.

3. For the CRISPR mutant it's stated that there is a lack of protein expression with a 'data not shown'. Since this data is available it would be worth adding as a supplemental.

Response 3: These mGluR6a knock-out fish are now published (<https://doi.org/10.1167/iavs.65.12.44>) and the immuno-histochemical results shows the lack of mGluR6a protein expression in mutants are presented as a supplemental figure of this study. We now better indicated this study when presenting these mutants in our methods and results section.

4. The resolution of the habenula synaptic connectivity does not allow for a rigorous or convincing conclusion and better images or membrane labeling would help solidify this important result. Notably, this is challenging with the utilized tools as so many versions of red fluorescence fluorophores are notorious for aggregation.

Response 4: Our results showed presence of presynaptic markers (in green-channel using synaptophysin-GFP) largely in the neuropil of Hb neurons dendrites (Figure 5I-K) and axon terminals, but not along the axon bundles of Hb neurons along fasciculus retroflexus (Suppl Figure 7A). Since these dendrites of Hb neurons are largely overlapping (at the core of Hb) it is not possible to see individual dendrites of Hb neurons by means of light microscopy. We, however, obtained better quality confocal images and updated our results in our revised Figure 5I-K. We hope that these new confocal data of better quality is convincing for visualization of Synaptophysin-GFP expression in dHb neurons.

Minor:

1. Can latency of excitatory and inhibitory responses be quantified from any of the current data, such as in Figure 1 or 5. This analysis could provide support from within this current study for points made in the discussion regarding the rates of excitatory and inhibitory responses.

Response 5: Calcium imaging data obtained at 2,5 Hz as we did in this study, would not be sufficient to make strong conclusions/claims on the latency of responses. Therefore, we choose not to include such claims in our results section, but we briefly mentioned this possibility in the discussion. We now removed those discussion on temporal dynamics or the latency from our discussion.

2. Figure 1A the z-axis is hard to appreciate. Providing two orthogonal views may provide more context

Response 6: We now provided two orthogonal views as asked.

3. Whenever average calcium traces are shown it would be helpful to show SEM for a reader to better appreciate the range of observed responses.

Response 7: We now added the SEM to all calcium traces, please note that in some places SEM might be too small to clearly see it.

4. 2nd sentence in abstract is confusing

Response 8: We rewrote this sentence, we hope this is clearer now.

5. Spatial transcriptomics (is this the molecular cartography?) could be better described in the methods and include the N's performed

Response 9: Yes, spatial transcriptomics is molecular cartography (brand name). We will rewrite this sentence. We have used Hb sections from n = 2 fish with identical results. The protocol for generation and source of data are now better described in the methods section

6. Providing some labeling of positions/regions in the spatial transcriptomics image would be helpful

Response 10: We now added schematic illustrations on the right of spatial transcriptome data (Figure 2C) marking where the sections are taken and the direction of sectioning. We hope this helps clarifying the images.

7. In Figure 2E middle panel are the different treatment conditions performed sequentially on the same individual. The line graph format gives this impression and should be clarified.

Response 11: Yes, these experiments were performed in the same animal, holding the same cell during different treatment conditions. We will better clarify this in the figure legends.

8. Figure 3B what is the grey area on the graph?

Response 12: Gray shaded area is the period that we used to analyze the effects of CPPG. We explained this now better in the figure legends.

9. In Figure 5D how many different habenula neurons were stimulated. Do you think number of responsive neurons would change with stronger stimulation of the target neuron?

Response 13: In each experiment in Figure 5 D-F, a single Hb neuron is micro-stimulated intracellularly, while imaging calcium responses of Hb neurons (n=28 neurons for control, n=18 for CPPG applied conditions). These numbers are now highlighted in figure legends. In fact, we were very excited to see that stimulating a single Hb neurons has a small but significant effect on the population of Hb neurons and it was sufficiently strong to make a point on interactions between Hb neurons and the role of group III mGluRs on these interactions. It is likely that stronger stimulation of Hb neurons would lead to stronger effects on the target neurons. Yet, we choose to perform our stimulations to reach the calcium activity levels comparable to spontaneous burst that we have observed earlier in our *in vivo* recordings of Hb (see

<https://doi.org/10.1016/j.cub.2021.08.021> Figure 1 and Figure 2B) , and in sensory-evoked activity in our present Figure 1. Hence, performing additional intracellular recordings/stimulation of Hb neurons (that are technically very challenging) will not add substantially to our current study or change our findings. We hope that the reviewer and the editors would agree with this.

10. It could be interesting to see if pharmacologic manipulation of G-protein ai function would recapitulate direct receptor manipulation. I don't think this experiment needs to be done yet may be worth considering/discussing.

Response 14: Further pharmacological dissections of mGluR cascade would be perfect for a future study. We now added this in the discussions.

Reviewer #2 (Remarks to the Author):

The study by Ostenrath et al. investigates the inhibitory action of Glutamate receptors, specifically the group III mGluRs, in the physiology of the zebrafish habenula and behavior. The authors provides important insights into the so-far understudied function of group III mGluRs and show their importance not only in the inhibitory neuronal responses to sensory stimuli, but also in animal behavior. It underscores the diverse action of Glutamatergic transmission through this group of mGluRs. The experiments are rigorously performed and the results are presented in a well-organized manner. On the other hand, my major concern is about an uncertainty about whether the group III mGluRs-dependent inhibition is directly mediated within the habenula (Hb) circuit or indirectly mediated by non-Hb neurons (as detailed below). This issue is mainly caused by the lack of specificity of pharmacological and genetic manipulation of the mGluRs. Anatomical analyses suggest some potential neural circuit inside the Hb, however, this anatomical finding is not integrated with the functional observations. Overall, the authors' interpretation that this inhibition is mediated through circuits within the Hb is not sufficiently supported by data in this manuscript. Another main concern is that the behavior tests that they employed to show defects in defensive behavior in mGluR6a have not been shown to require Hb activity, so the link between the mGluR6a, Hb and the behavioral effect is rather weak.

Major points:

1) Pharmacological manipulation of the group III mGluRs are performed by either bath application (juvenile) or ventricular injection (larvae) of antagonist (CPPG) and agonist (L-AP4). Both approaches act not only on the group III mGluRs expressed in the Hb, but also on those expressed anywhere in the brain. The authors show that the group III mGluRs are strongly expressed in the juvenile dorsal Hb compared to the forebrain (Fig. 2C,D), but no evidence is provided for whether they are expressed in different parts of the brain other than the Hb and forebrain. Previous paper (Huang et al., PLOS One, 2012) showed that mGluR6a is strongly expressed in the Hb, as well as other brain areas (such as optic tectum and midbrain-hindbrain-boundary) at larval stages, suggesting that CPPG can act on neurons outside the Hb. In line with this, CPPG application increases activity of Hb and olfactory bulb, as the authors describe, but

an area posterior to the Hb seem to show some increased activity upon CPPG administration, suggesting that neurons located outside of the Hb can be manipulated by CPPG (Fig. 3A). Therefore, one cannot exclude the possibility that the pharmacological effect is primarily mediated by non-Hb neurons, which in turn affect Hb neurons. Such indirect effects should be carefully discussed. Although the authors do acknowledge this by saying that “This cross-modal inhibition in Hb is by multi-synaptic inhibition through Hb-interpeduncular nucleus (IPN) connections onto presynaptic terminals of Hb neurons (115)” as a possibility, the overall tone of the manuscript is that the site of group III mGluRs’ action is on the Hb. For instance, “Our results revealed prominent inhibitory connections across Hb neurons, which are largely reduced by blocking group III mGluRs” (Introduction) and “We revealed that glutamatergic Hb neurons can directly communicate with each other” (Discussion).

Response 15: We thank the Reviewer 2 for highlighting the importance of our findings on the role of group III mGluRs in animal behavior and neural computations in Hb. We agree that group III mGluRs are expressed in multiple brain regions as we have presented in our forebrain spatial transcriptome data, as well as past studies by Haug et al 2013 (J. Comp Neuro) as well as Huang et al. 2012 (PLOS One) by using *in situ* labelling across the zebrafish brain. In all this past work and our current work, habenula seems to be at an intersection point for the combined expression of all group III mGluRs. Perhaps that’s also the reason why we see such strong impact of pharmacological perturbations of group III mGluRs (both agonist and antagonist) primarily in Hb.

While it is possible that small number of neurons elsewhere in the brain are affected from group III mGluRs pharmacology, they do not seem to be significant. To address this point, we now analyzed the data from mid-brain neurons in animals treated with CPPG and presented them in two new figures. When CPPG is bath applied in brain explants, we observed no significant activation of midbrain neurons above chance levels (Supplemental Figure 3). When CPPG is injected *in vivo*, we observed neither a significant amplification of midbrain sensory responses, nor a decrease in midbrain sensory selectivity (Supplemental Figure 5). We hope that these new results provide better support for the specific action of our pharmacological perturbation of group III mGluRs, and highlight Hb as an important site of action during these experiments.

We sincerely thank for this suggestion that allowed us to strengthen our argument and to focus our study primarily on Hb. Upon group III perturbation, we showed altered excitability, interactions and sensory computations in Hb neurons, in addition to altered defensive and adaptive behaviors. Yet, as the Reviewer 2 mentioned, we did not exclude the possibility that Hb neurons also interact through Hb-IPN connections. Therefore, we have presented these parallel and perhaps complementary pathways in our discussions. We also expanded our discussion to highlight the multi-synaptic interactions between Hb neurons.

Finally, to relate our results with CPPG injection (which is discussed above) and mGluR6a mutants, we also performed new behavioral experiments with juvenile zebrafish after CPPG injection. These new results with CPPG-injected fish showed identical behavioral alterations as in mGluR6a mutant fish. We hope that these additional behavioral experiments better link our results with perturbing group III mGluRs pharmacologically and genetically. Such extensive intersection of findings highlights the importance of group III mGluRs in regulating Hb computations and animal behavior. Further investigations on the role of mGluR driven inhibition in other brain regions, while appealing, would be the focus of another future study.

On a last note, we now adapted the tone of few sentences marked by the reviewer as below:

Our results revealed prominent inhibitory interactions across Hb neurons, when single Hb neurons are micro-stimulated.

We revealed that glutamatergic Hb neurons can interact with each other, largely through inhibition, which is sensitive to group III mGluR blockers.

2) In the example shown in Figure 5D, the electrically stimulated cells show extremely strong activity of DF/F more than 60. This does not seem to be physiological by comparing the Hb responses in Fig. 1A. Despite this strong activity in the simulated cell, the evoked activity in the Hb are rather small (around 2 DF/F). This could suggest that the connection between the simulated cells and the surrounding cells are rather weak or indirect. The authors should perform this experiment in more physiological conditions.

Response 16: In fact, we were very excited to see that stimulating even a single Hb neurons has a small but significant effect on the population of Hb neurons and it reveals interactions between Hb neurons (direct or poly-synaptic) and the role of group III mGluRs on these interactions. We choose to perform our stimulations to reach the calcium activity levels comparable to spontaneous burst that we have observed earlier in our *in vivo* recordings of Hb (see <https://doi.org/10.1016/j.cub.2021.08.021> Figure 1 and Figure 2B) , and in sensory-evoked activity in our present Figure 1. Due to these points, in fact, our stimulation intensity is at a level close to physiological activity levels of Hb neurons. We now highlight this in the method section, where we describe our micro-stimulation parameters. Performing additional intracellular recordings/stimulation of Hb neurons (that are technically very challenging) with different stimulation intensities will not add substantially to our current study or change our findings. We hope that the reviewer and the editors would agree with this.

3) The microstimulation experiment in Fig. 5D uses the whole brain explant. Related to my point 1), the evoked activity in the Hb could be indirectly generated by the Hb-IPN pathway. It would be interesting if the authors perform this experiment using the explants only containing the Hb, although this might be technically difficult.

Response 17: As we mentioned in **Response 16** our results on single cell microstimulation + calcium imaging in Hb revealed interactions between Hb neurons and that group III mGluRs play a role in these interactions. Our goal in this study is to investigate the role of group III mGluRs in Hb as close as it is possible to physiological and *in vivo* like conditions. This also includes an intact Hb-IPN connectivity. As we mentioned in **Response 15 and 16**, we now better highlighted that interactions between Hb neurons can be due to both direct and polysynaptic connections between Hb neurons and that group III mGluRs play a role in these interactions.

4) Interestingly, the authors describe that dHb neurons have presynaptic terminals in the Hb neuropil region (Fig. 5G, H). This is quite new and exciting, however I cannot see a direct innervation from the soma to this neuropil region from the image provided. Can they show a neurite that originate from cell bodies of dHb neurons that terminate in the Hb by looking at the tdTomato signal? This could be done by 3D projection or tracing of the neurites. In addition, can they show that the group III mGluRs are expressed near this terminal region? This would be a prerequisite for the glutamate released by dHb neurons to induce inhibition in the postsynaptic Hb neurons.

Response 18: Firstly, we do not argue that dHb neurons are the only source of glutamate in Hb. On the contrary, we believe that inputs coming from forebrain sensory-limbic systems in Hb are a major source of glutamate delivered to Hb neurons. We now expanded this in our discussion section.

Our results showed expression of presynaptic marker Synaptophysin-GFP in the neuropil of Hb neurons dendrites. Hence, we choose to share this observation and simply stated the expression of this presynaptic marker in the result text. If the Reviewer 2 and the editors prefer, we could drop our Synaptophysin-GFP images in dHb dendrites or move them to supplemental figures. However, as the Reviewer 2 pointed out this is an interesting observation that we think is worth sharing with the scientific community.

In our earlier version of the manuscript, the confocal images of Synaptophysin-GFP expression pattern were generated in fixed animals (to provide DAPI stain), hence the image quality of Synaptophysin-GFP expression was not very high. We now generated better quality images of Synaptophysin-GFP expression (Figure 5 I-J). As better presented in this new figure, the dendrites of multiple dHb neurons are largely overlapping at the core of Hb. Hence, even with this higher quality confocal image, it is not possible to see or reconstruct individual dendrites of dHb neurons (as requested by Reviewer 2). We should also add that we do not argue that Hb neurons dendrites are directly innervating cell bodies of Hb neurons. But instead, we observe that the dendrites of multiple Hb neurons appears to converge in the core of Hb, where they can interact with each other. Such dendro-dendritic synapses are observed in several brain regions (e.g. retina, olfactory bulb). Moreover, as we discussed some group III mGluRs are known to also be expressed at and act extrasynaptic locations (including mGluR6 expression on the dendrites Nicoletti et al 2011 (<https://doi.org/10.1016/j.neuropharm.2010.10.022>)). Therefore, we do not argue that direct connections between Hb neurons are a required for group III mGluR dependent interactions, but these interactions can operate extrasynaptically. We now indicated this possibility better in our discussion section and adjusted our tone when presenting these results.

5) It would be nice if they can show that the group III mGluRs-dependent inhibition in the Hb with the knowledge of Hb anatomical innervation pattern. For example, if they microstimulate one of the dHb neurons labeled in Tg(narp:GAL4VP16), do they observe the inhibition in putative postsynaptic cells located close to the Tg(narp:GAL4VP16)-positive neurites? This way they can correlate the spatial specificity of the intra-Hb projection and the evoked inhibition.

Response 18: To answer this point, we took two complementary approaches. Firstly, we have already showed that the impact of group III mGluR perturbations are significantly weaker in vHb (dao expressing neurons in Figure 3G) when compared to dHb neurons. To further investigate the anatomical location of group III mGluR pharmacological perturbation *in vivo*, we also reanalyzed our Hb data after splitting neurons in dorsal (top 40 μm) versus ventral (under top 40 μm) zones of Hb. In line with our earlier argument, we indeed observed that neurons in dorsal zones of Hb showed highly significant alterations in sensory computations, as compared to ventral neurons. These new results are now presented in Supplementary Figure 6 and integrated in the results.

Moreover, as suggested by the Reviewer 2, we further investigated the spatial distribution of excitatory and inhibitory interactions between Hb neurons upon micro-stimulation of individual Hb neurons. These results showed that Hb neurons that show excitation are significantly close to microstimulated neurons, as compared to inhibited Hb neurons (Figure 5 F). This is in line with the pairwise correlations of Hb neurons spontaneous activity as a function of distance

(Figure 5C), where nearby neurons exhibit positive correlations. These results also support that inhibitory interactions between Hb neurons are preferred between distant Hb neurons, which are likely a part of different functional Hb cluster/ensemble. These new findings are also integrated in the results section, and we hope that answers the comment of Reviewer 2 on the spatial distribution of interactions between Hb neurons.

6) Fig. 2 shows that the group III mGluRs are more strongly expressed in dHb than vHb at larval stage. In the CPPG experiments presented in Fig. 4 and 6, it is expected that the effect of CPPG is stronger for the dHb compared to the vHb. This can be tested by genetically labeling dHb or vHb neurons and repeating the same experiments.

Response 19: We agree that group III mGluR expression is mostly in dHb and less in vHb. In fact our results in Figure 3G indicate that group III mGluR pharmacology has significantly smaller impact on vHb neurons specifically labelled by *dao:Gal4*. However, a significant population of vHb neurons do still show alterations upon group III mGluR pharmacology (Figure 3G). This is somewhat expected as some group III mGluRs (mGluR7 and mGluR8) are also expressed in zebrafish vHb and rodent lHb (Figure 2A-C).

Yet, we still decided to test this suggestion from the Reviewer 2, by investigating differences on the effects of group III mGluRs perturbation on Hb sensory computations *in vivo*, between dorsal vs ventral zones. As described in **Response 18**, we reanalyzed our Hb data after splitting neurons in dorsal (top 40 μm) versus ventral (under top 40 μm) zones of Hb. We indeed observed that neurons in dorsal zones of Hb showed highly significant alterations in sensory computations of CPPG injected animals, as compared to ventral neurons. These new results are now presented in Supplementary Figure 6 and integrated in the results.

We are confident that these new results (Supplemental Figure 6) fully address the comment of the Reviewers 2 on the anatomical location of the action zone of group III mGluR perturbation. Our findings highlight dorsal Hb as the primary action area that is in line with our other findings with genetic labelling of vHb (Figure 3G) and gene expression analysis (Figure 2A-C).

7) The mutant model of mGluR6a is lacking mGluR6a protein in any parts of the brain. The authors cannot conclude that the observed behavioral phenotypes are caused by the Hb, unless they present evidence that the mGluR6a is not expressed anywhere in the body. As I mentioned in point 1) above, the mGluR6a is expressed not only in the Hb, but also in the optic tectum and midbrain-hindbrain boundary, suggesting that the Hb-independent effect cannot be excluded. The authors may be aware of this issue, but please explain them more explicitly.

Response 20: Our results indicate that Hb (especially dHb) have a special place as a group III mGluRs action zone, especially given the intersection of expression for multiple group III mGluRs in dHb. It is possible that relatively weak mGluR6a expression in midbrain reported in Haug et al 2013 (<https://doi.org/10.1002/cne.23240>) can have some impact on these behaviors. However, this is very unlikely, especially given our new data. We took this comment very seriously, and we did our best to provide new evidence ruling out this possibility, as explained below.

As described in **Response 15**, while it is possible that small number of neurons elsewhere in the brain are affected from group III mGluRs pharmacology, they do not seem to be significantly altered by our CPPG treatment. To address this point, we now analyzed the data from midbrain neurons in animals treated with CPPG and presented them in two new figures. When CPPG is

bath applied in brain explants, we observed no significant activation of midbrain neurons above chance levels (Supplemental Figure 3). When CPPG is micro-injected to forebrain *in vivo*, we observed neither a significant amplification of midbrain sensory responses, nor a decrease in midbrain sensory selectivity (Supplemental Figure 5). We hope that these new results provide better support for the relatively specific action of our pharmacological perturbation of group III mGluRs, and highlight Hb as an important site of action during these experiments.

Additionally, we performed additional experiments to relate our results with forebrain CPPG micro-injection (which is discussed above) and mGluR6a mutants. To do so, we micro-injected CPPG in zebrafish forebrain, and repeated some of the behavioral experiments that we previously have done with only mGluR6a mutants. These new results with CPPG micro-injected fish (with no obvious neural alterations in midbrain, Supplemental Figure 3 and 5) showed identical behavioral alterations as in mGluR6a mutant fish (Figure 7 E-H). We hope that these additional behavioral experiments better link our results with perturbing group III mGluRs pharmacologically and genetically. Such extensive intersection of behavioral alterations in CPPG forebrain injected fish and mGluR6a mutants highlights the importance of group III mGluRs in regulating Hb computations and animal behavior.

Further investigations on the role of mGluR driven inhibition in other brain regions, while appealing, would be the focus of another future study.

8) The authors state that “Previous research has shown that Hb is involved in various defensive behaviors (57-64, 66, 92, 101-103)”. This set of references includes studies from both mouse and zebrafish. It is unclear why they chose to test behavior that are different from those that have been already shown to involve the Hb. The behavior tests that they examined in this study generally address fish response to aversive stimuli, but none of them have been shown to require Hb activity. From the data, they can only conclude that the mGluR6a gene is involved in particular types of defensive behavior, but it is inconclusive whether or not the effect of mGluR6a is mediated by its expression in the Hb.

Response 21: We hope that our **Response 20** is convincing that both CPPG injection in the forebrain, which leads to Hb specific neural alterations (Supplemental Figure 3 and 5) and mGluR6a mutant exhibit identical alterations in animal behavior (Figure 7 E-H). This intersection of overlapping behavioral alterations narrows the effect of group III mGluRs to Hb.

The role of habenula in adaptive and defensive behaviors have been investigated across species both in rodents and in fish (Mouse: Lazaridis et al 2019, Baker et al 2017 (<https://doi.org/10.1016/j.neuroscience.2016.02.010>) (<https://doi.org/10.1038/s41380-019-0369-5>), Fish: Andalman et al 2019 (<https://doi.org/10.1016/j.cell.2019.02.037>), Amo et al 2014 (<https://doi.org/10.1016/j.neuron.2014.10.035>), Duboue et al 2017 (<https://doi.org/10.1016/j.cub.2017.06.017>). Moreover, data from patients of mood disorder (e.g. anxiety) and depression highlight an increase in Hb activity in human subjects (Sartorius et al 2007 (<https://www.sciencedirect.com/science/article/pii/S0306987707002472>), Lawson et al 2016 (<https://doi.org/10.1038/mp.2016.81>). In fact, our genetic and pharmacological perturbations targeting group III mGluRs resembles such Hb hyperexcitability. Therefore, we choose to perform assays that cover several aspects of animals’ anxiety-like defensive and adaptive behaviors. Novel tank diving behavior that we used, has been a classical assay to test anxiety and adaptive behaviors (Fontana & Parker 2022 (<https://doi.org/10.1016/j.jneumeth.2022.109706>), Fontana et al. 2022 (<https://doi.org/10.1007/s00213-021-05990-w>)), which we observed alterations in mGluR6a

mutants. Similarly light-dark transition assay has been used to test anxiety and adaptive behaviors fish (Fontana et al. 2022 (<https://doi.org/10.1007/s00213-021-05990-w>) Johnson et al. 2023 (<https://doi.org/10.1038/s41598-023-29668-9>), during which we observed that both pharmacological and genetic perturbation of group III mGluRs have revealed identical results (Figure 7 E-H). In addition to these, we also choose to perform an assay measuring animals' responses to mechanical vibration, which is typically an aversive stimuli eliciting fast escape followed by a long lasting bottom dive, similar to aversive diving responses elicited by alarm odors (Speedie and Gerlai et al. 2009 (<https://doi.org/10.1016/j.bbr.2007.10.031>), Masuda et al. 2024 (<https://doi.org/10.1016/j.cub.2024.02.003>). We observed that these defensive diving responses recover faster in control animals, when compared to mGluR6a mutants (Figure 7C-D). This mechanical vibration stimulus was an important way for us to link our neural data with animal behavior. Therefore, it is essential to present this result using these diverse behaviors.

It is possible that perturbing group III mGluRs may have an effect on some of the other behaviors that were used to study the impact of “ablating or silencing” Hb neurons, which is different than amplifying the sensory responses of dHb neurons as we do here (both by pharmacological and genetic perturbation of mGluRs). Therefore, we choose to employ these three different behavioral assays focusing on defensive and adaptive behavior. We now adjusted the result text to better explain our reasoning for choosing these assays.

We hope that our clarification is sufficient, as performing more and more various behavioral test would not change our conclusions on the role of group III mGluRs on Hb computations and animal behaviors.

Minor points

1) The authors' group previously showed that Hb responds asymmetrically to the light and odor stimuli (e.g. Fore et al., Science Advances, 2020). The Fig. 1A does not show clear asymmetry as previously reported. Why is it the case? Also, in this figure, there are different shades of orange and blue dots, but the explanations are not given. Please clarify.

Response 22: In our previous work (Dreostie et al 2014, Current Bioogy) we have shown strong asymmetries in odor and light responses in 5 day old zebrafish larvae. Our later work (Fore et al., Science Advances, 2020), showed that these asymmetries are highly reduced in 3 weeks old juvenile zebrafish. Therefore, the example figure in Figure 1A is in line with this observation of reduced asymmetries. Different shades of blue/orange dots are due to 3D visualization of these dots, some of which appear behind transparent grey dots. We now adjusted this figure for clarification and showed multiple orthogonal views.

2) On page 4, “Notably, we identified topographically distinct expression of mGluR4, mGluR6a, mGluR8b in the zebrafish dHb (Figure 3C)”: do they mean Fig. 2C, not Fig. 3C?

Response 23: Apologies this Figure 2C, this is now corrected.

3) Fig. 3D, G: the y-axis label should be “affected” but not “effected”.

Response 24: Thank you, this is now corrected.

4) In Supplementary Fig. 4, they checked whether the mGlu6a mutants recapitulate the

pharmacological alterations. Although they mention that “These results are very similar to the changes in sensory representations we observed upon blocking group III mGluRs via ventricular CPPG injections (Figure 4)”, but this statement is not correct because some phenotypes are not actually recapitulated and also they compare juvenile vs larvae. The authors should describe this more carefully and precisely.

Response 25: That’s true that CPPG injections were done in larval fish, and CPPG bath application was done in juvenile zebrafish brain explant, since this is what our ethical permission allows us to do. We explained in each result whether the experiments were performed in larvae and juvenile fish. As presented in Haug et al 2013 (<https://doi.org/10.1002/cne.23240>), the expression patterns of group III mGluRs are very similar between these ages, and hence our results are comparable. We now added this statement, when larval results are presented.

Moreover, we now performed additional behavioral experiments highlighting the overlap of results in CPPG micro-injected and mGluR6a mutant fish.

5) Fig. 7 legend: the authors might want to use “light and dark transition”, but not “dark and light transition”.

Response 26: thank you, this is done.

6) Fig. 7E: There are two scale bars for time. Which one is correct?

Response 27: This was a mistake, now corrected.

7) In the Discussion, they discuss conservation between zebrafish and mouse. Since very little conservation was found in terms of the expression patterns in this study (Fig. 2), this discussion seems not relevant in this case.

Response 28: Hasikawa et al. 2020 (<https://doi.org/10.1016/j.neuron.2020.03.011>) shows convincing level of conservation of Hb neurons across zebrafish, therefore it is important to discuss our results from the perspective across species. Moreover, both species expresses group III mGluRs in Hb (also see new data on mGluR8 in rodent mHb, Supplemental Figure 2 D-E, in line with zebrafish single cell and spatial transcriptomes in Figure 2), hence we argue that it is important to discuss this possibility. We now made the necessary adjustments to highlight this in the discussion section.

8) The discussion about the time scale of mGluR6a seems unrelated to the data presented in this manuscript.

Response 29: We dropped this from the discussion

Reviewer #3 (Remarks to the Author):

Reviewer #4 (Remarks to the Author):

Overview:

This manuscript addresses how glutamatergic signaling, mediated by metabotropic glutamate receptors (mGluRs), can affect the neural physiology, circuit function, and the behavior. Specifically, the authors use the habenula of larval and juvenile zebrafish to explore the roles of group III metabotropic glutamate receptors.

They first show that diverse sensory inputs can drive both excitation and inhibition in habenula neurons. There are more excited neurons than inhibited neurons for vibration and light stimuli, while there are more inhibited neurons in response to odors. After finding that group III mGluRs are expressed in the habenula, they show that an agonist hyperpolarizes Hb neurons in general, and that an antagonist depolarizes them. This supports the notion that glutamate, acting through group III mGluRs, has a generally inhibitory effect in the habenula. Calcium imaging in explants suggests that this hyper- and de-polarization has the effect in increasing and decreasing habenular activity, respectively. Moving back in vivo, they show that Hb neurons have greater excitation and less inhibition with to visual and vibration stimuli when mGluRs are blocked, with the knock-on effect that more neurons show responses (above 2SD) to both types of stimuli. This is followed by an analysis of correlations between pairs of Hb neurons at a range of distances, with and without antagonist, from which the authors draw the conclusion that mGluRs are responsible for inhibition that is manifested as negative correlations at longer distances. Such inhibition could support competition between sensory stimuli, so the authors then look at the (mostly suppressing) interactions between responses to combined visual and vibration stimuli. Antagonist shifted these effects subtly away from suppression between stimuli in the responses of Hb neurons. Finally, they showed that mutants for one of the group III mGluRs have similar activity patterns to those in fish treated with antagonist, and that these animals have enhanced defensive behavior.

Overall, I find the work interesting. It provides a satisfying example glutamatergic inhibition, different elements of this signaling through electrophysiology, calcium imaging, and behavior. I do not view the outcomes as groundbreaking, but think that they will be of interest to a wide audience of neuroscientists, principally those interested in glutamate and those interested in the role of the habenula. I have presented several major comments below. The most important of these is captured in comments 3 and 4. I am concerned that the generally increased excitability of Hb neurons treated with the group III mGluR antagonist explains most or all of the results, and that inhibition is not convincingly implicated by these analyses as currently presented.

Major comments:

#1

I find the evidence for mGluR expression in the mouse habenula to be weak. It is based on the analysis of a published dataset, and identifies robust transcript for only two out of four receptors that were studied. The abundance of these transcripts in the Hb versus other brain

regions is not described, and the relevance of the expression level values is not put in context. Overall, these data are much weaker than the equivalent zebrafish data. I assume that these results are included to generalize the conclusions drawn from the zebrafish work, but I believe that they lower the standard of the paper's data, and I would suggest removing them.

Response 30: We thank the Reviewer 3-4 for praising the quality of our work. We believe that it is important to put forward available rodent data and associated results regarding group III mGluRs. Therefore, in this revision we decided to include additional and newly released data sets from Allen Institute, which is in line with our earlier scRNAseq results that are analyzed and presented in our Figure 2. These new data sets include spatial transcriptome and imputed spatial gene expression patterns for several available group III mGluR genes (mGluR4, mGluR7 and mGluR8) in mouse habenula. This new data further support the expression for mGluR4, mGluR7 and mGluR8 (Supplemental Figure 2D-E). Unfortunately, mGluR6 is not a part of this gene imputation, so we could not investigate that gene. You may further inspect these imputed data here: ABC Atlas , and spatial transcriptome data for mGluR8, here: ABC Atlas). Moreover, these data sets shows that several other mouse brain regions (as in fish, Figure 2D), also express group III mGluRs. We already highlighted this broad expression pattern of group III mGluRs in mouse brain in our discussion and now added an additional sentence regarding this important question. Hence it is likely that our work, focusing primarily in the habenula, will initiate further investigations for the function of group III mGluRs in brain function. We hope that the Reviewer 3-4 and the editors will agree with us and support our decision to keep our scRNAseq results (in Figure 2B) and extend further support our findings by the new spatial gene expression data (Supplemental Figure 2D-E). We argue that keeping these rodent gene expression results are a valuable addition to our study.

#2

The nature of the sensory stimuli applied in Figure 1 is important. Odor, in particular (although vision also), can be either attractive or threatening, and without testing a range of stimuli, it is difficult to gauge whether all odors lead principally to inhibition or whether different odors lead to different balances of excitation and inhibition. This becomes an important consideration when behavioral responses enter the story.

Response 31: In Figure 1, we presented our observation that sensory evoked inhibition is common in Hb circuits, across sensory modalities. However, we choose not to use odor stimuli further in our calcium imaging and behavioral experiments during pharmacological and genetic perturbation of group III mGluRs. This is primarily due to strong expression of group III mGluRs in the olfactory bulb (Supplemental Figure 1). Such an expression would not allow us to disentangle whether the effect we observe in Hb is due to the function of group III mGluRs in Hb or in the olfactory bulbs. This was also a major reason why we chose to focus on only on visual and vibrational stimuli in the rest of our study. Studying the impact of group III mGluRs in the olfactory computations and the olfactory bulb is better suited for a future study.

Nevertheless, we still took the advice of the Reviewer 3-4 and performed additional experiments and presented our results with 3 different odor categories (Supplemental Figure 1). As it is seen in these results the odor evoked inhibition can be found in the habenula across different odor categories. We prefer not to include these results in our manuscript, which do not add substantially to our findings, while slightly defocusing our study. But if the Reviewer 3-4 and the editors recommend us to keep this new Supplemental Figure 1, we will do so.

In general, insufficient technical detail is provided about the sensory stimuli. What is the lux change in the imaging chamber when the red LED is illuminated? Where is the solenoid located and what is the volume of the vibration stimulus within the chamber? Did you try a range of stimuli before deciding that these were the appropriate ones for testing individually and a multisensory combinations? Since rather careful analyses of responses to both visual and vibrational stimuli exist in the literature, this information is important for putting the current results in context.

Response 32: Our stimulation parameters are described in the methods section with specific parameters to control the specific stimulation devices. Hence, all our stimulation conditions are reproducible given these parameters. These stimuli were selected based on the specifications that we have used in our earlier studies (Dreosti et al 2014, Current Biology, Fore et al 2020 Science Advances). We now highlighted these studies in our methods section better. Our stimuli were chosen by testing multiple intensities to evoke robust responses without generating stimulation artifacts in our calcium imaging data.

#3

The claims made around the data in Figure 4 need more consideration. It is convincing that there is more activation by stimuli in Hb neurons when the antagonist is present. This effect spans visual and vibration stimuli. Therefore, it is unsurprising (maybe inevitable) that more neurons will exceed 2SD with each stimulus. This is presented as evidence that Hb neurons become less selective to stimuli, when all that has been shown is that they are more excitable. The claim regarding selectivity depends on there being something sacred about 2SD, which there is not.

Perhaps a claim of less selectivity could be made if the ratio of visual to vibrational stimuli changed (to become more similar to one another with antagonist), but this would have the caveat that the calcium signal is not a linear representation of activity. All things considered, I do not buy the interpretation that they are less selective; I am only convinced that they are more excitable.

Response 33: Yes, in the presence of group III mGluR perturbations, we observed that Hb neurons are disinhibited which give rise to less inhibition. Hence, we observe more excitation in Hb. We agree that less selective responses above 2SD is a consistent result in a hyperexcitable system. We also agree that 2SD is not a sacred value, but it is a cut-off that is consistently defined for identifying significant changes, hence we choose to keep this commonly-used and simple-to-communicate threshold to define, whether a neuron is responding to only one of our stimuli, to both of our stimuli or to none of them.

Nevertheless, we would like to note that we did not only use this measure of multi-modal neurons when defining alterations in response selectivity. We also presented correlations between sensory representations (Figure 4I, Supplemental Figure 8K). Such correlations between sensory representations across hundreds of Hb neurons, encompass encoding of sensory stimuli by population vectors representing the activity of all Hb neurons, and not only the ones responding above 2SD threshold. Such response correlations are a very commonly-used method to quantify how selective/similar stimulus representations are. Hence, we believe that our response vector correlation analysis presented in Figure 4I and Supplemental Figure 8K addresses the concern raised by the Reviewer 3-4 and shows that visual and vibrational

responses become more similar to one another with both antagonist and with mGluR6a mutation. We believe that our description for the use of correlations in order to quantify similarities of sensory representations is concisely explained in the present text.

#4

As with the argument about selectivity in Figure 4, the claims about inhibition and distance presented in Figure 5 may be overlooking a simpler explanation. If Hb neurons are simply more responsive with CPPG, then positive correlations may increase since synchronized ensembles might be larger or stronger. I do not like how the data in 5E and F are presented given this possibility. Given that the blue data in E and the grey data in F are the same datapoints, I would prefer to see this treatment presented only once. The current display and description give the false impression that they are data from different experiments. Then, I would like to see both excitation and inhibition for both aCSF and CPPG (four bars). This will clarify for the reader that a single experiment has been done, and will distinguish between the specific loss of inhibition in CPPG versus a generally more excitable (and correlated) network. The excitation data from the CPPG treatment are needed to make this judgement.

Response 34: Yes, we agree that increased correlation in Figure 5C is due to disinhibition elicited with CPPG increasing neural activity and as a result increasing the correlations. We now highlighted this in the results text.

About the presentation of our electrophysiology results in Figure 5: Yes, the blue dots in Figure 5E and grey dots in Figure H are the same data points. We now added a sentence explaining this in the figure legends better. However, these data (in Figure 5E-F and Figure 5G-H) are presented for different purposes and for different comparisons and hence, we choose to keep these panels separately. Figure 5E panel reveals that upon micro-stimulating a single Hb neurons, we observe primarily inhibition in significantly larger fraction of surrounding Hb neurons. This was not known before and important to report. Later in Figure 5F (a new panel that is added during this revision), we showed that those inhibited Hb neurons are significantly further away from the micro-stimulated neurons, when compared to (fewer) excited neurons. Such a spatial organization of interactions between Hb neurons is also a novel finding and it is important to report.

Figure 5G and Figure 5H on the other hand, now moves to a new concept explaining the role of group III mGluR blockers on Hb connectivity. Therefore, it is important to keep these presentations separately. Figure 5G is a newly added figure panel, showing that group III mGluR blockers has no significant impact on the excitatory interactions between Hb neurons, as requested by Reviewer 3-4. Figure 5H complements Figure 5G and reveal that group III mGluR blockers significantly reduce the ratio of inhibited neurons.

We believe that presenting our results in the way we delivered is extremely important to clarify these different observations and novel findings. We hope the editors and the reviewers will agree with us.

Minor comments:

#5

In Figure 2 E and F, is the experimental n given the number of neurons or the number of explants? If neurons, what was the number of explants?

Response 35: This is the number of explant/fish and neurons, since each explant can be used only once per drug experiment. We clarified this better in the figure legends.

#6

I'd like to see a panel like 3B for L-AP4 so that the timecourse and magnitude of the agonist can also be appreciated.

Response 36: We now provided this in the revised Supplemental Figure 3

#7

In Figure 3, was co-labeling with DAO really necessary to identify the vHb? I had thought that there was going to be in interpretation of what role DAO+ neurons might specifically be doing, but if they're just to show which neurons are ventral, the 3D image should have been sufficient to show this. I feel like the DAO angle is a little distracting.

Response 37: Here we use dao:Gal4 as a specific marker to label vHb neurons. Therefore, we choose to present the results benefiting from the genetic specificity of these experiments. We believe that presenting the results with dao labelled vHb neurons are complementary to our transcriptome data, and important to observe that not all subregions of Hb are equally affected by group III mGluR perturbation. We believe that these results are important to report, and we hope that Review 3-4 and the editors would agree with us to keep them as a part of this study.

Moreover, inspired by the comment of Review 3-4, we performed additional analysis for the effects of CPPG injection across dorsal ventral axis of Hb (Supplemental Figure 6). These new results revealed that sensory responses of neurons located in dorsal zones of Hb are more susceptible to pharmacological perturbation of group III mGluRs, when compared to minimal alterations in neurons located in ventral zones of Hb.

Dear Editors and Reviewers,

We are delighted to see the reviewers' enthusiasm for our study and greatly appreciate their compliments on the quality of our data and results. We are also grateful for their constructive and thoughtful feedback, which has certainly helped improve the clarity and value of our study.

After carefully considering all the comments, we have prepared a revised manuscript that addresses them to the best of our ability. You will find our detailed responses below, where we explain how we addressed each comment through new experiments, analyses, clarifications, textual adjustments, adapted figure panels and new supplemental figures. To help with the revision, we highlighted the updated parts of our manuscript using **yellow highlights**. We are confident that our manuscript is now significantly strengthened and more comprehensive. We thank all reviewers for their suggestions.

All reviewers' comments are included below in **red**, and our responses are provided in black.

We look forward to hearing your feedback on our revised manuscript.

With kind regards,

Emre Yaksi

Anna Maria Ostenrath

Reviewer #1 (Remarks to the Author):

The authors have sufficiently addressed my concerns and comments. This is a compelling story supported by some very nice experiments.

Response 1: We thank the Reviewer 1 for all the constructive feedback.

Reviewer #2's

The authors have adequately addressed most of my questions, by adding new data and revising the text. However, I would like to highlight three unresolved points that should be addressed before publication.

Response 2: We thank the Reviewer 2 for all the constructive feedback. We did our best to answer additional points by new experiments and analysis as explained below.

-Re. Response 18: Thank you for addressing this point. The new images provided in Fig. 5 now better illustrate the presynaptic marker in the neuropil of the Hb. In the authors' response, they discuss the possibility that inhibition may be mediated between Hb dendrites, possibly by the dendro-dendritic synapses. Although this is speculative at this stage, it would be valuable to explicitly mention this interesting possibility in the manuscript, and I encourage the authors to do so. I feel that the images in Fig. 5G deserve further explanation.

Response 3: We thank for this suggestion. We are indeed excited about this possibility of dendro-dendritic synapses. Since our results hints this possibility, we took the suggestion of the reviewer #2 and added 2 sentences about the possibility of dendro-dendritic synapses in the 2nd paragraph of our discussion section.

-Re. Response 21: I agree that the behavior assays that they performed are well-established for testing defensive and adaptive behaviors. However, to my knowledge, the behavior assays that the authors used here have not been shown to be mediated by the Hb (Bühler and Carl, 2021, <https://doi.org/10.3390/biom11020324>). For example, silencing dHb in zebrafish has been shown not to alter performance in the novel tank diving test (Chou et al., 2016, DOI: 10.1126/science.aac9508). Therefore, I still think that it is unclear whether the behavioral defects in the mGluR6a mutants are due to alterations in Hb function. The new data in Fig 7G nicely link the site of action of the group III mGluRs. Can they test whether the CPPG injection also phenocopies the other behavioral tests (novel tank assay and mechanical vibration)? These data would be valuable, as it could further support the connection between the group III mGluRs, Hb and the behavior.

Response 4: We agree with the point made by Reviewer 2 regarding silencing dHB that was shown to not alter novel tank diving performance (and findings of Chou et al., 2016, DOI: 10.1126/science.aac9508). However, in our experiments, we do not silence dHB. On the contrary, we provide several evidence that our pharmacological and genetic group III mGluR perturbations increase Hb responses. We appreciate the suggestion to inject CPPG and perform the novel-tank assay and mechanical vibrations in these animals compared to the controls. To inject CPPG into the forebrain ventricles, animals need to be briefly anesthetized with MS222. This was a prerequisite set by our ethical committee for these injections. Several colleagues we consulted advised against performing the classical novel-tank diving test after animals were anesthetized with MS222, as such MS222 treatment would impact some of the results. Hence, we chose not to conduct this novel tank diving test in CPPG-injected animals. Instead, to address this suggestion, we adopted an alternative strategy and present a similar experiment (now shown in Supplemental Figure 10 A-B), where CPPG-injected animals are placed in a novel behavioral arena viewed from the top, allowing us to measure animals swim speed. In these new results, consistent with the novel-tank diving test, we observed that control animals initially reduce their swim speed upon placement in the new arena (Supplemental Figure 10 A-B) and later gradually adapt and increase their swim speed within the first 10 minutes. Our new results revealed that control animals adapt and

increase their speed significantly more than CPPG-injected animals. We hope these new results complement our earlier novel-tank diving results in mGluR6a mutants and sufficiently address Reviewer 2's comments.

Reviewer 2 also requested data on the vibration response of freely swimming CPPG-injected fish. The experiments mentioned above also allowed us to test the altered swim speed change in response to vibrations in CPPG-injected fish, as requested by Reviewer 2. Since animals are viewed from the top in this relatively shallow behavioral arena, we analyzed altered swim speed in response to vibrations. In these new results, we observed that upon mechanical vibrations, both control and CPPG-injected zebrafish initially and transiently increase their swim speed transiently. However, CPPG-injected animals showed a significantly stronger reduction in swim speed during the delayed phase of the vibration response (Supplemental Figure 10 C-D), resembling our vibration results with mGluR6a mutants. We hope these new results sufficiently address Reviewer 2's comments and strengthen our arguments regarding alterations of adaptive and defensive behaviors in CPPG-injected animals. These experiments and analysis are now incorporated in our results section.

-Re. Response 28: The new data on the spatial transcriptomics of mouse mGluR8 (presented in Supplementary Figure 2D, E) is somewhat confusing. The data here shows that mouse mGluR8 is expressed more strongly in the medial Hb (MHb), while the data in Figure 2B indicates that the expression of mouse mGluR8 is stronger in the lateral Hb (LHb). How can this discrepancy be reconciled?

Response 5: Indeed, spatial transcriptome results are more sensitive for detecting GPCRs, and hence we were very excited when the Allen Institute released their new data with group III mGluRs. mGluR8 is the only group III mGluR presented in the Allen Institute's mouse resources with direct MERFISH multiplexed in situ hybridization results. Therefore, we are confident that the spatial transcriptome results with mGluR8 expression in the mouse mHb are very solid, also in line with our zebrafish mGluR8 expression in dHb (Figure 2C), and hence we included these results (Supplemental Figure 2D-E). The single-cell RNAseq results presented in our Figure 2B for the mouse also come from the Allen Institute's mouse data, and expression levels for mGluR4 and mGluR7 are well captured in LHb, which is in line with the Allen Institute's spatial gene imputation results (in Supplemental Figure 2D). We believe the discrepancy between single-cell and spatial mGluR8 results is due to the very low capture rate for mGluR8 transcripts in single-cell RNAseq, similar to the mGluR6 gene with a low capture rate (Figure 2B). While spatial transcriptome data is superior (when available), we still believe that it is best to present all available data sets for all available genes for type 3 mGluR expression in mice, for consistency and transparency. Hence, if Reviewer 2 agrees, we prefer to keep the presentation of these results as it is. We hope that our clarification was sufficient to convince Reviewer 2 about our choice to present these results in this form.

Reviewer #4 (Remarks to the Author):

There are improvements in this manuscript, but still some important concerns that

have not been addressed. These are detailed below, according to the numbers of the original comments.

Major comment 1:

I think that the addition of new data makes the mouse study less cursory, but I still believe that its inclusion detracts from the paper's focus and does not support an argument for a key role for the habenula (given the broad expression for the three mGluR genes studied). While I continue to disagree about the value of these data, I don't see them as an obstacle to publication.

Response 6: We are glad to hear that the reviewer 4 think that the addition of new data makes the mouse study less cursory, and do not find our mouse single cell and spatial gene expression results as an obstacle to publication. We will keep these results presented in our study, as they are.

Major comment 2:

I see the new supplementary figure 1 as a valuable addition. It can be mentioned quickly in the text, so as not to interfere with the manuscript's focus, but still addresses the concern about the valence of olfactory cues. I would suggest leaving it in the manuscript's supplementary section.

Response 7: We will keep this as supplemental figure as suggested.

My concern about stimulus parameters has not been addressed.

I still believe that some actual measurements of the stimuli are in order. Saying the wattage or voltage across the LEDs or solenoid does not tell the reader what the animal is experiencing. Citing past work does not solve this problem unless measurements are reported in those papers.

Response 8: Our lab and other labs have published several papers where the stimulus parameters are presented as we have done in our current manuscript. It is indeed very difficult to know precisely what a small animal in an agarose gel is experiencing, when we present our stimuli. Yet, it is important to present the stimulus parameters as precisely as we can, so it is reproducible across labs and experimenters. Hence, we did some additional measurements and presented these in our methods and even results sections. For the ambient light-flash stimulation measurements, we added the precise LED spectral parameters to show what spectrums of light are detected, in this ambient flash of light. We also placed the photometer device exactly at the location of the fish during the experiment and recorded and reported the light amplitude in watts at the 630 nm spectrum range matching our LED. We do not have any other way to better report the ambient light level increase that is presented to our animals. We believe this is sufficiently well description of a light flash stimuli, on par with similar studies in top journals. For the mechanical vibration stimulation, we have recorded the amplitude and frequency of the vibration stimulation, which is now presented in Supplemental Figure 3D. We are thankful for this suggestion, which clearly shows that the primary frequencies the animal experiences are above 200 Hz, even stronger above 1000Hz, meaning that primarily the auditory system is activated by our transient (50 ms) mechanical vibration. We now tried to better clarify the parameters of our stimuli to the best of our ability. With our current technological abilities, we are not able to provide

further information to better specify our stimulus parameters. We hope that this updated information sufficiently addresses this comment, and our stimulus descriptions are acceptable for publication.

Major comment 3:

This comment has not been meaningfully addressed.

Given that 2SD is arbitrary (if frequently used), it would be good to show results from a few different thresholds to show that the results being reported are general, rather than being dependent on the use of a specific threshold.

Response 9: As requested, we now present these results with different STD thresholds, showing no differences in our findings and arguments (especially on the concept of increased multi-modality in CPPG-injected animals) across thresholds (Supplemental Figure 4). We believe this makes a solid point that our findings are not merely an effect of the threshold. We believe that the addition of these iterative analyses is convincing enough regarding our conclusions on increased multi-modality and hence less selective responses in CPPG-injected animals.

The threshold used is less important than the interpretation, though. A general increase in the population will both have the effect of producing more multimodality (at a given threshold) and also the effect of increasing the correlations across treatments. If more neurons across the Hb are active with antagonist or in the mutant, and if that activity is tightly correlated to stimulus presentations, it will result in higher correlation values, even if there is no biologically meaningful lack of selectivity taking place. All of these effects could be the results of a simple increase in responsiveness.

Response 10: In hundreds of papers investigating sensory selectivity across multi-neuronal response patterns, Pearson's correlations have been used. Hence, we respectfully disagree with the idea that an increase in Pearson's correlations do not represent a meaningful lack of selectivity, especially in light of our new iterations with multi-modality using different thresholds. All our analyses point towards a habenula (Hb) where significantly more Hb neurons exhibit responses to more than one stimulus in CPPG-injected animals. We agree that increased responsiveness and multi-modality (hence less selective) are parameters that are changing in an increasingly excitable (hence increasingly responsive) Hb in animals with group III mGluR perturbations. Therefore, we are unsure why Reviewer 4 questions the validity of our findings and interpretations, particularly in regard to the statement “no biologically meaningful lack of selectivity is taking place”. We could think of only one additional analysis that can be added to our manuscript to complement Pearson's correlations in measuring the similarity of sensory representations, which is cosine similarity. Cosine similarity is different from Pearson's correlations, as it calculates the angle between two response vectors, and the amplitude of the vectors is not relevant for the cosine similarity measurements. Our new results (Supplemental Figure 4J) showed that CPPG-injected animals exhibit significantly higher cosine similarity compared to controls, indicating that sensory representations are less selective between light and vibration responses.

In Figure 4H and Supplementary Figure 8J, it appears that the unimodal and multimodal populations add up to 100%. As such, it is unnecessary (I would say inadvertently misleading) to represent them as separate data, especially with the inclusion of apparently different statistical tests on the same data. I would suggest presenting these as stacked bar graphs to avoid confusion.

Response 11: We updated this in all relevant figures.